# ACCELERATING TRAINING WITH NEURON INTERACTION AND NOWCASTING NETWORKS

**Boris Knyazev**[1] **Abhinav Moudgil**[2,4] **Guillaume Lajoie**[3,4]
**Eugene Belilovsky**[2,4] **Simon Lacoste-Julien**[1,3,4]

[1]Samsung – SAIT AI Lab, Montreal  [2]Concordia University  [3]Université de Montréal  [4]Mila
b.knyazev@samsung.com

https://github.com/SamsungSAILMontreal/nino

## ABSTRACT

Neural network training can be accelerated when a learnable update rule is used *in lieu of* classic adaptive optimizers (e.g. Adam). However, learnable update rules can be costly and unstable to train and use. Recently, Jang et al. (2023) proposed a simpler approach to accelerate training based on weight nowcaster networks (WNNs). In their approach, Adam is used for most of the optimization steps and periodically, *only every few steps*, a WNN nowcasts (predicts near future) parameters. We improve WNNs by proposing neuron interaction and nowcasting (NiNo) networks. In contrast to WNNs, NiNo leverages neuron connectivity and graph neural networks to more accurately nowcast parameters. We further show that in some networks, such as Transformers, modeling neuron connectivity accurately is challenging. We address this and other limitations, which allows NiNo to accelerate Adam training by up to 50% in vision and language tasks.

## 1 INTRODUCTION

Modern deep learning models, such as large language models (Touvron et al., 2023), are trained using classic adaptive optimizers, such as Adam[1] (Kingma & Ba, 2015; Loshchilov & Hutter, 2017). These optimizers update neural network parameters $\theta \in \mathbb{R}^n$ using gradient descent at step $t$ as $\theta_{t+1}^i = \theta_t^i - \Delta\theta_t^i, \forall i \in [1, n]$, where $\Delta\theta_t^i$ is the update computed analytically based on the history of parameter values, gradients and the learning rate. Recently, Jang et al. (2023); Sinha et al. (2017) showed that parameters $\theta$ follow a predictable trend so that optimization can be accelerated by **nowcasting** (predicting near future) parameters using a learnable function $f^\phi$: $\hat{\theta}_{t+k}^i = \theta_t^i + f^\phi(\theta_t^i, \theta_{t-1}^i, \theta_{t-2}^i, ...)$, where $k \gg 1$ is the future horizon. More popular learnable

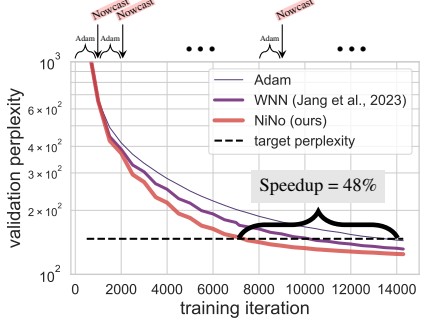

Figure 1: Adam without and with nowcasting using our NiNo vs WNN (Jang et al., 2023) on a language task that NiNo and WNN have not seen during their training.

approaches to speed up optimization, such as "learning to optimize" (L2O), are recurrently applied at every step $t$ (Andrychowicz et al., 2016; Metz et al., 2022b). Compared to L2O, the parameter nowcaster $f^\phi$ is applied **very rarely** reducing its overhead, e.g. a base optimizer such as Adam is run for 1k steps followed by the prediction step (Fig. 1). Moreover, training such $f^\phi$ is simpler than L2O, since a supervised loss can be used instead of a more challenging meta-learning loss with recurrent inner steps. However, akin to Adam, $f^\phi$ from prior works does not directly leverage the structural information of $\theta$, such as connectivity between neurons and layers. This structure has been shown to be critical for many parameter representation tasks, such as property prediction (Navon et al., 2023; Zhou et al., 2024b; Kofinas et al., 2024).

---

[1]We use Adam throughout the paper, but our discussion and methods are in principle applicable to any optimizers that produce a trajectory of parameters, including SGD with/without momentum, AdamW, Adagrad, etc.

In this work, we propose **neuron interaction and nowcasting (NiNo)** networks making better predictions of future parameters to **accelerate training with a base optimizer such as Adam** and make the following contributions:

1. We introduce NiNo, which directly uses the structure of neural network parameters by leveraging recently proposed *neural graphs* (graphs of neurons) (Kofinas et al., 2024; Lim et al., 2024).

2. We improve Transformer's neural graphs of Kofinas et al.; Lim et al. by more accurately modeling the *neuron permutation symmetry* of multi-head self-attention. Our neural graphs combined with graph neural networks create a strong inductive bias to make rare but accurate predictions.

3. NiNo is conditioned on the future horizon $k$ to allow for prediction in the near and far future without retraining it, facilitating its usage in diverse tasks and at different optimization stages.

4. We demonstrate that NiNo accelerates training with Adam for ConvNets and Transformers reducing the number of steps to achieve the target performance of Adam by up to 50%. We release our source code and models at `https://github.com/SamsungSAILMontreal/nino`.

## 2 RELATED WORK

**Learning to optimize (L2O).** The L2O literature has offered many approaches to learn a neural network (*optimizer*) that optimizes the parameters of other neural nets (*optimizees*) (Andrychowicz et al., 2016; Chen et al., 2022b;a; Amos, 2022). Among them, Safeguarded L2O (Heaton et al., 2023; Prémont-Schwarz et al., 2022) is most related to ours as it switches between an L2O optimizer and SGD/Adam. While Safeguarded L2O alleviates the *meta-generalization* challenge (Thérien et al., 2024), training an L2O optimizer remains costly and unstable due to the use of meta learning methods and its long inner loop unrolls required at each meta-training iteration (Metz et al., 2019; 2022b). Moreover, overheads of L2O add up at each iteration making it more computationally intensive than Adam. In contrast, our approach follows *weight nowcaster networks* (WNNs) (Jang et al., 2023), where the nowcaster model is applied very rarely (e.g. once per 1k steps of Adam) making the total overhead negligible, yet still speeding up training significantly.

**Parameter prediction and generation.** This area has been active recently, primarily aiming at reducing computational costs of training neural nets (Peebles et al., 2022; Ashkenazi et al., 2023; Schürholt et al., 2022; 2024; Knyazev et al., 2021b; 2023; Zhou et al., 2024d; Soro et al., 2024; Wang et al., 2024). Most related to our work are IntrospectionMLP (Sinha et al., 2017) and WNNs (Jang et al., 2023) serving the basis of our approach. These methods train simple MLPs to periodically (e.g. every few epochs of Adam) predict future parameter values of a neural net given its past parameters (Section 3.1). However, their MLPs predict parameter coordinate-wise (for each parameter independently) similar to optimization methods without leveraging the structure of neural networks. Moreover, their MLPs predict parameters only for a predefined future horizon $k$, whereas different tasks and different optimization stages can have different parameter evolution trends (Morchdi et al., 2022; Guille-Escuret et al., 2024; Lange et al., 2023). We address these shortcomings by conditioning the prediction on $k$.

**Representation learning of neural network parameters.** This area has also developed fast recently (Navon et al., 2023; Schürholt et al., 2021; 2024; Ashkenazi et al., 2023; Andreis et al., 2023; Zhou et al., 2024b;c;a). One of the main goals in these works is to model *neuron permutation symmetry* – the property of neural networks that if the neurons in one layer are permuted in the same way as the neurons in the next layer, the neural network preserves its function (Hecht-Nielsen, 1990). Accurate modeling of this symmetry allows for better estimation of network properties or encoding implicit neural representations. To model neuron permutation symmetry, Kofinas et al. (2024); Lim et al. (2024) proposed *neural graphs* (graphs of neurons) enabling the usage of graph neural networks (GNNs) (Kipf & Welling, 2017; Corso et al., 2020). Remarkably, as GNNs can digest graphs of any size and connectivity, the synergy of neural graphs and GNNs enables processing diverse parameters from different architectures and tasks using a *single* GNN. We leverage both the neural graphs and GNNs to learn a single NiNo model that can accelerate optimization in diverse tasks. However, the neural graphs of Kofinas et al. (2024); Lim et al. (2024) do not accurately model neuron permutation symmetry of Transformers (Section 4.1). We address that shortcoming to make better predictions by NiNo.

**Dynamic models.** Our approach of learning from the interaction of nodes in a neural graph to make future predictions is also related to dynamic interactive models, where message passing networks and GNNs are used to learn the interaction within relational temporal networks (Trivedi et al., 2019; Knyazev et al., 2021a) and physical systems (Kipf et al., 2018; Sanchez-Gonzalez et al., 2018; 2020;

Zambaldi et al., 2019). However, developing a strong parameter prediction model requires many specific design choices and careful modeling of neuron permutation symmetry, motivating our work.

## 3 BACKGROUND

We consider a neural network parameterized by $\theta_t \in \mathbb{R}^n$ trained for $t$ steps with Adam (or another optimizer). Our goal is to accelerate optimization by predicting future parameters $k \gg 1$ steps ahead given its past values using a meta-network $f_k^\phi$, parameterized by $\phi$: $\hat{\theta}_{t+k} = \theta_t + f_k^\phi(\theta_t, \theta_{t-1}, \theta_{t-2}, ...)$.

### 3.1 WEIGHT NOWCASTER NETWORKS (WNNS)

WNNs (Jang et al., 2023) model $f_k^\phi$ as an MLP predicting the delta (update difference) of the $i$-th parameter: $\Delta\hat{\theta}_{\tau+k}^i = f_k^\phi(\tilde{\theta}_\tau^i, \tilde{\theta}_{\tau-1}^i, ..., \tilde{\theta}_{\tau_c}^i)$, where $\tau_c = \tau - c + 1$ and $c$ is the context length. Here, $\tau, \tau - 1, ...$ are **epoch indices** and $\tilde{\theta}^i$ are the parameters *scaled* based on the range of values in $\theta_\tau^i, ..., \theta_{\tau_c}^i$ (see details in Section A.1). The predicted parameter value is obtained by unscaling the predicted delta: $\hat{\theta}_{\tau+k}^i = \theta_\tau^i + \text{unscale}(\Delta\hat{\theta}_{\tau+k}^i)$. WNNs are trained in a supervised way by collecting a training dataset of parameter trajectories $\{[\theta_1, \theta_2, ...]\}_1^C$ obtained with Adam and applying the $l_1$-loss:

$$\text{argmin}_\phi ||\Delta\tilde{\theta}_{\tau+k} - \Delta\hat{\theta}_{\tau+k}||_1, \tag{1}$$

where $\Delta\tilde{\theta}_{\tau+k}$ are the scaled target parameter deltas. Sinha et al. (2017) proposed the approach of future parameter prediction originally, but without scaling the parameters and without a simple way to choose at which optimization steps to make the prediction. In contrast, WNNs scale the parameters and are always applied **periodically**, once per every $c$ epochs of a base optimizer (Adam) with $k = c$, which better accelerates optimization. For example, to use a trained WNN $f_k^\phi$ on a new task, first Adam is run for $c$ epochs ($c = 5$ by default) after which $f_k^\phi$ is applied to predict future (10-th epoch) parameters, after which the procedure repeats (the base optimizer is continued for another 5 epochs followed by parameter prediction) until convergence. So the WNN is applied very rarely, e.g. in the case of 200 Adam steps per epoch, the WNN is applied only once per 1,000 steps (Fig. 1).

### 3.2 (NAIVE) NEURAL GRAPH OF TRANSFORMERS

WNNs apply an MLP for parameter $\theta^i$ given only its past values. Such an MLP is inherently limited, since future parameter values depend on many factors, including connectivity of neurons. Modeling neuron connectivity in a way that generalizes to arbitrary network structures to make parameter prediction as general as possible is challenging. Simple parameter representations, e.g. flattening of parameters into a vector (Schürholt et al., 2021), are not general and do not model *neuron permutation symmetry* (the neurons can be permuted without changing the network output). Recently, Kofinas et al. (2024); Lim et al. (2024) proposed to represent $\theta$ using a *neural graph* $\mathcal{G}(\theta) = (\mathbf{V}, \mathbf{E})$ with node features $\mathbf{V} \in \mathbb{R}^{|\mathbf{V}| \times d_\mathbf{V}}$ and edge features $\mathbf{E} \in \mathbb{R}^{|\mathbf{V}| \times |\mathbf{V}| \times d_\mathbf{E}}$, where $d_\mathbf{V}, d_\mathbf{E}$ are the node and edge feature dimensionalities, respectively. For example, Equation 2 shows $\mathbf{E}$ for the $d_\mathbf{E} = 1$ dimensional edge features of an $L$-layer MLP (we ignore biases for simplicity): $\theta = \{\mathbf{W}^{(1)}, ..., \mathbf{W}^{(L)})\}$, where $\mathbf{W}^{(l)} \in \mathbb{R}^{d_{l-1} \times d_l}$ are the weights for $l \in [1, L]$.

Neural graphs can theoretically represent the parameters of any neural network taking into account neuron permutation symmetry as the rows and columns of weight matrices correspond to the nodes in the neural graph. This way, neural graphs impose a strong inductive bias similarly to using convolution for images. So a model operating on neural graphs, such as a graph neural network (GNN) and our NiNo model, can be used in diverse tasks and should be able to learn parameter prediction rules that generalize better with fewer samples than, for example, MLPs. However, to fully leverage the neural graphs in practice, we need to accurately construct them, which is not trivial for such networks as Transformers, motivating our approach in this work.

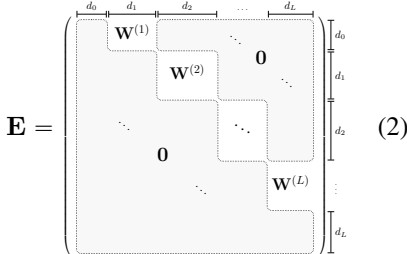

$$\mathbf{E} = \begin{pmatrix} \mathbf{W}^{(1)} & & & \\ & \mathbf{W}^{(2)} & & \mathbf{0} \\ & & \ddots & \\ & \mathbf{0} & & \ddots \\ & & & \mathbf{W}^{(L)} \end{pmatrix} \tag{2}$$

**Transformers.** For input $\mathbf{x} \in \mathbb{R}^{N \times d}$, where $N$ is a sequence length, a multi-head self-attention (MSA) layer (Vaswani et al., 2017) for each head $h \in [1, H]$ projects $\mathbf{x}$ using $\mathbf{W}_h^q, \mathbf{W}_h^k, \mathbf{W}_h^v$ followed

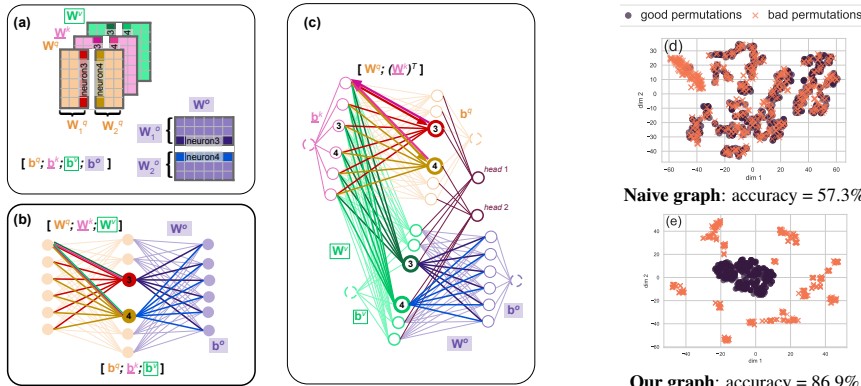

Figure 2: (a) MSA weights with $d = 6$ and $H = 2$ heads; (b) its naive (Kofinas et al., 2024) and (c) our neural graph with color-coded edge/parameter types. (d-e) Neural graph embeddings obtained using a GNN and projected to 2d using tSNE for 1k **good** and **bad** permutations of the MSA weights: (d) naive graph, (e) our graph. Accuracy is computed by fitting logistic regression for neural graph embeddings with labels being if the MSA output changes or not as described in Section A.6.

by self-attention and projection using another weight matrix $\mathbf{W}_h^o$ (see Section A.4 for details):

$$\mathbf{A}_h = \frac{(\mathbf{x}\mathbf{W}_h^q)(\mathbf{x}\mathbf{W}_h^k)^T}{\sqrt{d}} \quad \text{and} \quad \mathbf{y}_h = \text{softmax}(\mathbf{A}_h)(\mathbf{x}\mathbf{W}_h^v) \quad \text{and} \quad \text{MSA}(\mathbf{x}) = \sum_{h=1}^{H} \mathbf{y}_h\mathbf{W}_h^o. \quad (3)$$

To construct neural graphs of MSA, Kofinas et al. (2024); Lim et al. (2024) use $d_\mathbf{E} = 3$ dimensional edge features $\mathbf{e}_{ij} = \left( \left(\mathbf{W}^q\right)_{ij}, \left(\mathbf{W}^k\right)_{ij}, \left(\mathbf{W}^v\right)_{ij} \right)$, assuming that the multi-head case is automatically handled by the neural graph. However, as shown in Section 4.1, this neural graph does not model the neuron permutation symmetry correctly, therefore we refer to it as a *naive neural graph* (Fig. 2a,b).

## 4 METHODS

In this section, we describe the details of our neural graphs for an MSA layer of Transformers and the neuron interaction and nowcasting (NiNo) networks operating on such graphs (Fig. 4). Our neural graphs can be constructed for different Transformer architectures, including GPT2 (Radford et al., 2019), Llama (Dubey et al., 2024) (see graph examples in Fig. 9) and others (see our implementation). Following WNNs, NiNo is also applied very rarely during the optimization process (Section 3.1). However, NiNo leverages the representation power of neural graphs and GNNs to speed up optimization more significantly than WNNs, as we show in Section 5.

### 4.1 NEURAL GRAPH OF TRANSFORMERS

To construct neural graphs that accurately model neuron permutation symmetry in MSA, we (1) restrict neuron permutations across heads and (2) relax permutations of weights $\mathbf{W}^q, \mathbf{W}^k$ (Fig. 2).

**(1) Restricting permutations across heads.** We first observe that splitting computations into $H$ parallel heads (Equation 3) breaks neuron permutation symmetry across the heads, so shuffling neurons across the heads may change the overall output of MSA($\mathbf{x}$). Consider, for example, an MSA layer with $d = 6$ and $H = 2$ (Fig. 2a) and the dot product between outputs $\mathbf{x}_1^q$ and $(\mathbf{x}_1^k)^T$ obtained after projecting $\mathbf{x}$ using the weights of the first head $\mathbf{W}_1^q$ and $\mathbf{W}_1^k$: $\mathbf{x}_1^q(\mathbf{x}_1^k)^T = [\mathbf{x}_1^q, \mathbf{x}_2^q, \mathbf{x}_3^q][\mathbf{x}_1^k, \mathbf{x}_2^k, \mathbf{x}_3^k]^T$. The result of this dot product does not change when permuting neurons in both $\mathbf{W}_1^q$ and $\mathbf{W}_1^k$ using the same $\pi$, since it results in permuting the outputs: $\pi(\mathbf{x}_1^q)\pi(\mathbf{x}_1^k)^T = \mathbf{x}_1^q(\mathbf{x}_1^k)^T$. However, consider shuffling neurons across head 1 and 2 in $\mathbf{W}^q$ and $\mathbf{W}^k$, e.g. neurons 3 and 4 (highlighted in different colors in Fig. 2a) resulting in: $[\mathbf{x}_1^q, \mathbf{x}_2^q, \mathbf{x}_4^q][\mathbf{x}_1^k, \mathbf{x}_2^k, \mathbf{x}_4^k]^T$. Now the dot product no longer equals $\mathbf{x}_1^q(\mathbf{x}_1^k)^T$ unless $\mathbf{x}_4^q\mathbf{x}_4^k = \mathbf{x}_3^q\mathbf{x}_3^k$. So self-attention matrix $\mathbf{A}_1$ for head 1 and MSA($\mathbf{x}$) can change.

To take into account this restriction, we allow for neuron permutations **only within a head** by adding a separate node for each head that connects the appropriate neurons so that permuting neurons 3 and 4 from our example changes the graph while permuting withing a head does not change it (Fig. 2c).

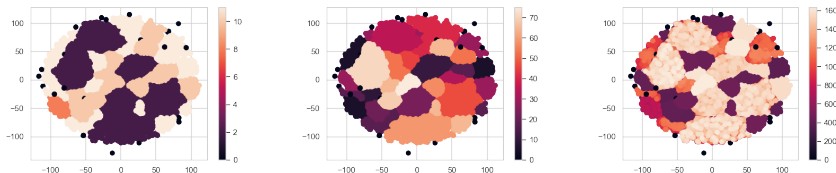

Figure 3: Laplacian Positional Encoding (LPE) of our neural graph's nodes color-coded by the layer type (left), layer index (middle) and neuron index (right) for a Transformer with 6 layers and 384 hidden units. We show tSNE projections from the 8-dimensional LPE to 2d.

**(2) Relaxing permutations for $\mathbf{W}^q, \mathbf{W}^k$.** We also observe that neurons in $\mathbf{W}_h^q$ and $\mathbf{W}_h^k$ can be permuted in a different way ($\pi'$) from $\mathbf{W}_h^v$ and $\mathbf{W}_h^o$ without changing the MSA output:

$$(\mathbf{x}\mathbf{W}_h^q)(\mathbf{x}\mathbf{W}_h^k)^T = (\mathbf{x}\pi'_c(\mathbf{W}_h^q))(\mathbf{x}\pi'_c(\mathbf{W}_h^k))^T \quad \text{and} \quad (\mathbf{x}\mathbf{W}_h^v)\mathbf{W}_h^o = (\mathbf{x}\pi_c(\mathbf{W}_h^v))\pi_r(\mathbf{W}_h^o), \quad (4)$$

where $\pi_c$ and $\pi_r$ denote permutation $\pi$ of the columns and rows in the matrix respectively. In the naive graph the neurons in $\mathbf{W}_h^q, \mathbf{W}_h^k, \mathbf{W}_h^v, \mathbf{W}_h^o$ are always permuted in the same way ($\pi_c = \pi'_c$) making the neural graph unnecessary restrictive (Fig. 2b). To address this issue, we keep $\mathbf{W}_h^q, \mathbf{W}_h^k, \mathbf{W}_h^v$ as **separate 1-dimensional edge features** in a neural graph instead of stacking them as 3-dimensional edge features (Fig. 2c). Since $\mathbf{W}_h^q$ and $\mathbf{W}_h^k$ share the input and output neurons while $\mathbf{W}_h^k$ is transposed in Equation 3, it is important to preserve the edge direction, so that for example $\mathbf{e}_{3,4}$ corresponds to the weights of $\mathbf{W}_h^q$ while $\mathbf{e}_{4,3}$ corresponds to the weights of $\mathbf{W}_h^k$.

## 4.2 Edge and Node Features

**Edge features.** In our neural graph, all the model parameters $\theta$ and auxiliary connections $\theta'$, such as residual and head connections introduced in Section 4.1, are represented using the edge features $\mathbf{E} \in \mathbb{Z}^{|\mathbf{V}| \times |\mathbf{V}| \times d_\mathbf{E}}$ so that $||\mathbf{E}||_0 = |\theta| + |\theta'|$ (Fig. 2c). To differentiate between $\theta$ and $\theta'$, we associate an integer edge type with each edge: $\mathbf{E}^{\text{type}} \in \mathbb{Z}^{|\mathbf{V}| \times |\mathbf{V}| \times 1}$, so our neural graph $\mathcal{G}(\theta) = (\mathbf{V}, \mathbf{E}^{\text{type}}, \mathbf{E})$.

**Node features.** For the node features we use the Laplacian Positional Encoding (LPE) (Belkin & Niyogi, 2003; Dwivedi et al., 2023) that allow graph neural networks to capture structural information more easily. For example the LPE implicitly embeds nodes of the same layer and same type close to each other even though this information is not explicitly provided (Fig. 3). To compute the LPE, we use *unweighted edges*, transform the graph to an *undirected* one and extract 8 smallest non-trivial eigenvectors, so our node features are $\mathbf{V}^{\text{lpe}} \in \mathbb{R}^{|\mathbf{V}| \times 8}$. In addition, for Transformer's word embedding layers, we found it beneficial to leverage a positional feature $\mathbf{V}^{\text{w}} \in \mathbb{Z}^{|\mathbf{V}| \times 1}$, since these layers often have the same size with ordered neurons. For other layers, this feature is set to zero.

**Sequence of parameters as a neural graph.** We consider a history of $c$ past parameter vectors following WNNs (Section 3.1): $\Theta_{\tau:\tau_c} = [\theta_\tau, \theta_{\tau-1}, ..., \theta_{\tau_c}] \in \mathbb{R}^{n \times c}$, where $\tau_c = \tau - c + 1$ as in Section 3.1. Since each parameter vector in $\Theta_{\tau:\tau_c}$ represents the same neural network structure, we transform the entire $\Theta_{\tau:\tau_c}$ into a single neural graph $\mathcal{G}(\Theta_{\tau:\tau_c}) = (\mathbf{V}^{\text{lpe}}, \mathbf{V}^{\text{w}}, \mathbf{E}^{\text{type}}, \mathcal{E}_{\tau:\tau_c})$, where $\mathcal{E}_{\tau:\tau_c}$ is obtained by stacking edge features: $\mathcal{E}_{\tau:\tau_c} = [\mathbf{E}_\tau, \mathbf{E}_{\tau-1}, ..., \mathbf{E}_{\tau_c}] \in \mathbb{R}^{|\mathbf{V}| \times |\mathbf{V}| \times d_\mathbf{E} \times c}$, whereas node features $\mathbf{V}^{\text{lpe}}, \mathbf{V}^{\text{w}}$ and edge types $\mathbf{E}^{\text{type}}$ do not change over time.

## 4.3 Neuron Interaction and Nowcasting Networks (NiNo)

Given an input neural graph $\mathcal{G}(\Theta_{\tau:\tau_c})$, our NiNo model processes it using layerwise scaling layer, node and edge embedding layers and GNN layers with a hidden size $D$. NiNo then predicts edge features for multiple steps in the future $\hat{\mathcal{E}}_{1:K}$ that are mapped back to the parameter space $[\Delta\hat{\theta}_{\tau+1}, ..., \Delta\hat{\theta}_{\tau+K}]$ using the inverse neural graph construction $\Delta\hat{\theta}_{\tau+k} = \mathcal{G}^{-1}(\hat{\mathcal{E}}_k)$ (Fig. 4).

**Layerwise scaling.** Since parameter values can vary in scale significantly across architectures and tasks, it is important to scale them appropriately. Compared to WNNs (Jang et al., 2023) scaling each parameter independently using min-max, we extend a layerwise scaling (Schürholt et al., 2022) to a sequence of parameters. Specifically, given weights $\mathbf{W}_{\tau:\tau_c}^{(l)} \in \mathbb{R}^{d_{l-1} \times d_l \times c}$ of the $l$-th layer of the input $\mathcal{E}_{\tau:\tau_c}$, we obtain scaled weights as $\tilde{\mathbf{W}}_{\tau:\tau_c}^{(l)} = (\mathbf{W}_{\tau:\tau_c}^{(l)} - \mu_\tau^{(l)})/\sigma_\tau^{(l)}$, where $\mu_\tau^{(l)}, \sigma_\tau^{(l)}$ are the scalar mean and standard deviation of $\mathbf{W}_{\tau:\tau_c}^{(l)}$. After repeating this step $\forall l \in [1, L]$, we obtain scaled $\tilde{\mathcal{E}}_{\tau:\tau_c}$.

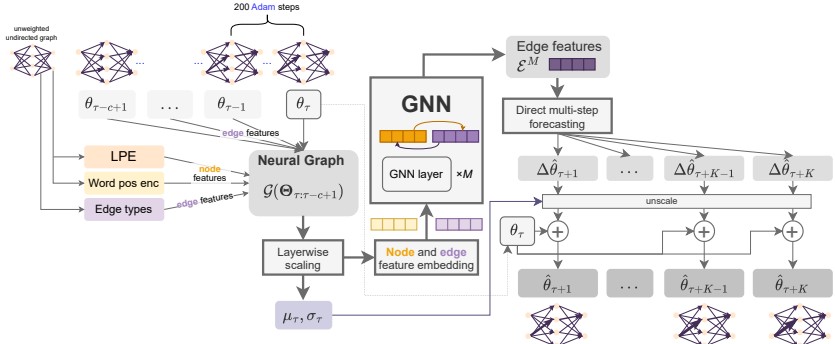

Figure 4: Our neuron interaction and nowcasting (NiNo) model. We feed $c$ past parameter states (**without gradients**) as input and predict future $K$ states leveraging our improved neural graph (Section 4.1) using a GNN. For a new optimization task, NiNo is applied rarely over time: only once per 1k steps of Adam (Section 5).

**Embedding layers and GNN.** We use linear $\phi_{\text{lpe}}, \phi_e$ and embedding $\phi_w, \phi_{\text{type}}$ layers to project node and edge features to the $D$-dimensional space followed by $M$ GNN layers and the output layer $\phi_{\text{DMS}}$:

$$\mathbf{V}^0 = \phi_{\text{lpe}}(\mathbf{V}^{\text{lpe}}) + \phi_w(\mathbf{V}^w), \qquad\qquad\qquad \mathbf{V}^0 \in \mathbb{R}^{|\mathbf{V}| \times D}, \quad (5)$$

$$\boldsymbol{\mathcal{E}}^0 = \phi_e(\tilde{\boldsymbol{\mathcal{E}}}) + \phi_{\text{type}}(\mathbf{E}^{\text{type}}), \qquad\qquad\qquad \boldsymbol{\mathcal{E}}^0 \in \mathbb{R}^{|\mathbf{V}| \times |\mathbf{V}| \times D}, \quad (6)$$

$$\mathbf{V}^m, \boldsymbol{\mathcal{E}}^m = \text{GNN}_{\phi_m}(\mathbf{V}^{m-1}, \boldsymbol{\mathcal{E}}^{m-1}), \qquad\qquad\qquad m = 1...M, \quad (7)$$

$$\hat{\boldsymbol{\mathcal{E}}} = \phi_{\text{DMS}}(\boldsymbol{\mathcal{E}}^M), \qquad\qquad\qquad \hat{\boldsymbol{\mathcal{E}}} \in \mathbb{R}^{|\mathbf{V}| \times |\mathbf{V}| \times K}. \quad (8)$$

Our GNN layers are based on Kofinas et al. (2024), but to enable better efficiency on large models, we use a simple mean aggregation. Specifically, given node $i, j$ features $\mathbf{v}_i^{m-1}, \mathbf{v}_j^{m-1}$ and edge features $\mathbf{e}_{ij}^{m-1}$, the $m$-th GNN layer is formulated as: $\mathbf{v}_i^m = \phi_a^m \left( \frac{1}{|\mathcal{N}(i)|} \sum_{j \in \mathcal{N}(i)} \mathbf{m}_{ij}^{m-1} \right)$, where $\mathbf{m}_{ij}^{m-1}$ is a message computed by aggregating features $\mathbf{v}_i^{m-1}, \mathbf{v}_j^{m-1}, \mathbf{e}_{ij}^{m-1}$; $\phi_a^m$ is an MLP; $\mathcal{N}(i)$ are the neighbors of node $i$; $\mathbf{e}_{ij}^m$ is then obtained using another MLP given $\mathbf{v}_i^m, \mathbf{v}_j^m, \mathbf{e}_{ij}^{m-1}$ (see Section A.5).

**Direct multi-step forecasting (DMS).** The final layer $\phi_{\text{DMS}}$ outputs $K$ values for each edge corresponding to future steps from $\tau + 1$ to $\tau + K$. This is motivated by direct multi-step forecasting performing well in time series forecasting by avoiding error accumulation effects of autoregressive forecasting (Chevillon, 2007; Zeng et al., 2023). To train the model given a training dataset of parameters trained with Adam, we use Equation 1, but applied for $k = [1, ..., K]$ instead of fixing $k$:

$$\text{argmin}_\phi \frac{1}{K} \sum_{k=1}^K (||\Delta \tilde{\boldsymbol{\theta}}_{\tau+k} - \Delta \hat{\boldsymbol{\theta}}_{\tau+k}||_1), \quad (9)$$

where $\Delta \hat{\boldsymbol{\theta}}_{\tau+k} = \mathcal{G}^{-1}(\hat{\boldsymbol{\mathcal{E}}}_k)$ and $\tilde{\boldsymbol{\theta}}_{\tau+k}$ are the target parameters scaled using $\mu_\tau, \sigma_\tau$. The predicted parameters are obtained by unscaling the predicted delta using $\mu_\tau, \sigma_\tau$: $\hat{\boldsymbol{\theta}}_{\tau+k} = \boldsymbol{\theta}_\tau + \text{unscale}(\Delta \hat{\boldsymbol{\theta}}_{\tau+k})$.

**Inference with $k$-decay.** Once our model is trained, it can predict future parameters for $k \in [1, K]$. While $k$ can be treated as a hyperparameter and kept fixed during optimization of the target task, we found that in the initial optimization stage using very large $k$ is beneficial, whereas in the later optimization stages $k$ should decrease fast, since the parameter values do not change significantly. Therefore, we propose to decay $k$ during optimization as $k \approx K((T - t)/T)^p$, where $T$ is the maximum number of optimization steps and $p$ controls the decay speed (Fig. 12). Although $p$ can be tuned, we found that $p = 2$ works well in our experiments.

## 5 EXPERIMENTS

We experiment with nine tasks, each defined by a dataset and a neural network architecture in the vision or language domains (Table 1). Four of the tasks, the *in-distribution tasks*, are of a relatively smaller scale and used to train our *meta-models* (NiNo, WNN and their variants). The other five tasks, the *out-of-distribution tasks*, differ from the in-distribution tasks in the architecture and/or dataset. We also experiment with larger GPT2-based and Llama-based architectures at the end of Section 5.2.

**Indexing parameters.** Since the notion of epoch used for $\tau$ in WNNs (Section 3.1) can be ill-defined in some tasks (e.g. in language tasks the models are often trained only for 1 epoch), in our experiments we found $\tau$ and $\tau + 1$ indexing parameters at step $t$ and $t + 200$ to be a well performing strategy in all the tasks, so for our default context length $c = 5$ we apply the meta-models every $5 \times 200 = 1,000$ steps.

Table 1: **In-distribution and out-of-distribution tasks.**

| | IN-DISTRIBUTION TASKS | | | | OUT-OF-DISTRIBUTION TASKS | | | | |
| | FM/16 | C10/16 | LM1B/3-24 | LM1B/2-32 | FM/32 | C10/32 | C100/32 | LM1B/3-64 | WIKI/3-64 |
|---|---|---|---|---|---|---|---|---|---|
| Final training loss | 0.25±0.06 | 0.91±0.1 | 5.94±0.03 | 5.85±0.04 | – | – | – | – | – |
| #models | 300 | 300 | 200 | 200 | – | – | – | – | – |
| #params | 14K | 15K | 1.2M | 1.6M | 56K | 57K | 63K | 3.4M | 3.4M |
| Target (validation) metric | Acc | Acc | Perplexity | Perplexity | Acc | Acc | Acc | Perplexity | Perplexity |
| Target value | 89.5% | 66.0% | 352 | 319 | 90.5% | 72.5% | 39% | 181 | 147 |
| Adam #steps | 8606 | 8732 | 23000 | 23500 | 8269 | 8607 | 8341 | 23500 | 13500 |
| NiNo #steps | 4582 | 3775 | 11500 | 12000 | 4395 | 4323 | 4646 | 12000 | 7000 |

Figure 5: Scaling (a,b) and generalization (c) trends for NiNo vs WNN+.

**Training and evaluation pipeline:**

1. **Meta-training:** Train the meta-models with the supervised parameter prediction loss (Equation 1 for WNN and Equation 9 for NiNo) using training checkpoints from the four *in-distribution tasks*.

2. **Usage:** Train task-specific neural networks in all the tasks using the Adam (vision) and AdamW (language) optimizers with the meta-model applied every 1k steps given $c = 5$ past parameters.

3. **Evaluation:** Define a target validation performance level for each task based on running the baseline Adam optimization (without the meta-model applied) and reporting the relative reduction of the number steps to achieve that performance by other methods.

## 5.1 SETUP

**Vision tasks.** We use the FashionMNIST (FM), CIFAR-10 (C10) and CIFAR-100 (C100) datasets and two convolutional architectures with three layers: with 16, 32 and 32 channels per layer (e.g. task FM/16) or 32, 64 and 64 channels per layer (e.g. task FM/32). The convolutional architectures are the same as in the L2O experiments of Kofinas et al. (2024) to enable fair comparison. In all cases, these tasks are optimized using Adam (Kingma & Ba, 2015) without weight decay, with a constant learning rate of 6e-3 (for FashionMNIST) or 3e-3 (for CIFAR) with a batch size of 128 for $T$=10k steps.

**Language tasks.** We use the LM1B (Chelba et al., 2013) and WikiText103 (WIKI) (Merity et al., 2016) datasets and train GPT2 style Transformers (Radford et al., 2019) with 3 layers, 24 hidden units and 3 attention heads (denoted as 3-24); 2 layers, 32 units and 2 heads (2-32) or 3 layers, 64 units and 4 heads (3-64). These tasks are optimized for the next token prediction loss with AdamW (Loshchilov & Hutter, 2017), weight decay 1e-2, learning rate 2e-4, batch size of 32, sequence length of 1024 for either 1 epoch (for LM1B) or 4 epochs (for WIKI) corresponding to around 24k or 14k steps respectively. We use a predefined GPT2 tokenizer in all the cases with a fixed vocabulary.

**Meta-training dataset.** We use FM/16, C10/16, LM1B/3-24 and LM1B/2-32 as the in-distribution tasks training 300, 300, 200 and 200 models in each task respectively ($C = 1000$ models in total). We save the checkpoints of intermediate steps, having in total around $1.6 \times 10^6$ checkpoints. Our dataset is large and diverse (Fig. 10), yet it is relatively cheap to be collected as the tasks are small.

**Baselines.** As a reference optimization method we use Adam for vision and AdamW for language tasks with a constant learning rate that we tune independently on each task based on the validation metric. Following WNNs (Jang et al., 2023) we use *Linefit* as another baseline (Section A.2). We further improve it with a simple scaling term to promote more recent parameters denoted as *Linefit+* (see Section A.3). As the strongest baseline, we use WNN with our layerwise scaling and $k$-decay which we denote as WNN+. Finally, we use the learning to optimize (L2O) model from Kofinas et al. (2024), which showed strong results on similar vision tasks. We use their L2O (NG-GNN) pretrained on FM/16 (denoted as L2O/FM16) as retraining their model on all our in-distribution tasks is expensive. For a fair comparison with L2O/FM16, we also trained NiNo on FM/16 only (denoted

Table 2: **Reduction (%) of the number of steps until a target performance w.r.t. Adam.** We observe that on both in-distribution and out-of-distribution tasks our models improve over the base optimizer and other acceleration methods.

| | IN-DISTRIBUTION TASKS | | | | OUT-OF-DISTRIBUTION TASKS | | | | | AVG SPEEDUP |
|---|---|---|---|---|---|---|---|---|---|---|
| | FM/16 | C10/16 | LM1B/3-24 | LM1B/2-32 | FM/32 | C10/32 | C100/32 | LM1B/3-64 | WIKI/3-64 | |
| Linefit | 27.0 | 26.8 | 32.6 | 31.9 | 16.6 | 26.5 | 20.7 | 29.8 | 33.3 | 27.3 |
| WNN | 29.3 | 31.4 | 26.1 | 23.4 | 22.2 | 27.9 | 25.0 | 27.7 | 25.9 | 26.5 |
| Linefit+ | 27.3 | 28.3 | 32.6 | 29.8 | 23.3 | 27.0 | 25.0 | 25.5 | 33.3 | 28.0 |
| WNN+ | 33.8 | 45.1 | 45.7 | 44.7 | 31.6 | 44.0 | 37.8 | 38.3 | 37.0 | 39.8 |
| NiNo (naive graph) | **48.7** | 53.1 | 33.7 | 12.8 | 42.1 | 49.3 | 37.8 | 23.4 | 33.3 | 37.1 |
| NiNo | 46.8 | **56.8** | **50.0** | **48.9** | **46.8** | 49.8 | **44.3** | **48.9** | **48.1** | **48.9** |

Table 3: **Ablations and analysis of hyperparameters. Average speedup (%) is reported.**

(**a**) Varying scaling

| | |
|---|---|
| layerwise ($\mu$, $\sigma$ per layer) | **48.9** |
| $\mu$, $\sigma$ per param | 41.5 |
| min-max per param | 38.8 |
| no scaling | 32.1 |

(**b**) Varying $p$ in $k$-decay

| | |
|---|---|
| $p = 1$ | 45.8 |
| $p = 2$ | **48.9** |
| $p = 10$ | 47.7 |
| $p = 0$ (no $k$-decay) | 40.8 |

(**c**) Ablating embeddings (Equation 5)

| | |
|---|---|
| LPE + word pos enc ($\mathbf{V}^{\mathrm{w}}$) | **48.9** |
| no LPE | 47.5 |
| no word pos enc ($\mathbf{V}^{\mathrm{w}}$) | 45.4 |
| no edge type | 45.7 |

(**d**) Varying context length

| | |
|---|---|
| $c = 3$ | 29.8 |
| $c = 5$ | **48.9** |
| $c = 7$ | 44.0 |
| $c = 10$ | 34.0 |

(**e**) Varying NiNo size

| | |
|---|---|
| $M = 3, D = 128$ | **48.9** |
| $M = 3, D = 32$ | 42.6 |
| $M = 1, D = 128$ | 39.7 |
| $M = 2, D = 128$ | 44.7 |

(**f**) Other hyperparameters ($b, \gamma$ are the batch size and learning rate used to meta-train NiNo)

| | |
|---|---|
| $C = 10^3, b = 4, \gamma = 3e\text{-}3$ | **48.9** |
| $C = 10^2, b = 4, \gamma = 3e\text{-}3$ | 45.1 |
| $C = 10^3, b = 4, \gamma = 1e\text{-}3$ | 41.5 |
| $C = 10^3, b = 2, \gamma = 3e\text{-}3$ | 43.8 |

as NiNo/FM16). We train WNN+ and NiNo with $M \in [1, 2, 3]$ layers and $D \in [16, 32, 64, 128, 256]$, where $M = 3, D = 128$ are used unless otherwise stated (see training details in Section A.7).

**Evaluation.** We follow Dahl et al. (2023) when choosing the approach to compare training methods and set our target based on the validation set performance using tuned Adam, which we consider as the main baseline (Table 1). To reduce evaluation uncertainty we train all the tasks multiple times (10 for vision and 3 for language) and use the median number of steps. We report a relative reduction of the number of steps in %, e.g. for task WIKI/3-64 if the method achieves perplexity 147 in 7000 steps (as our NiNo does), the reduction is 48.1% based on Table 1. For language tasks due to more expensive evaluation we only evaluate every 500 steps, whereas for vision we evaluate every step.

## 5.2 RESULTS

**Main results.** Our NiNo model shows consistently better performance than Linefit and WNN and their improved variants on all the nine tasks (Table 2). On average we speed up Adam optimization by 48.9% reducing the number of steps to achieve the target performance roughly by half. NiNo is followed by WNN+ and NiNo with a naive graph (Section 4.1). While the latter achieves slightly better performance on the FM/16 in-distribution task, in the out-of-distribution tasks this model significantly underperforms to NiNo, especially in the language tasks. This performance gap is expected since NiNo represents neural graphs for Transformers more accurately as we further validated qualitatively (Fig. 2d,e) and quantitatively (Section A.6). Other baselines perform poorly, but we highlight that both Linefit baselines compare favorably to WNN indicating that the latter could have learned a very simple rule despite training on a lot of data.

**Comparison with L2O.** L2O/FM16 performs the best in-distribution on the same task (Fig. 6a), however when applied to an unseen task it overfits to the training set (Fig. 6b). In contrast, our NiNo/FM16 trained on the same task as L2O/FM16 performs well on the validation set of unseen tasks, presumably because Adam is used for most of the steps and for the input NiNo uses the trajectory of parameters of the target task to make the prediction. In addition, L2O/FM16 and L2O in general have an overhead at every optimization step making the total computation cost noticeable (Table 7 in Section A.7). It is also more computationally challenging to meta-train it on multiple tasks, therefore we only use a pretrained L2O/FM16 from Kofinas et al. (2024). Since L2O/FM16 does not reach the target validation scores in most tasks, we compare it only using the curves in Fig. 6a,b and only on the FM/16 and C10/32 tasks.

**Ablations.** We ablate NiNo components and hyperparameters and found that our layerwise parameter scaling and $k$-decay are the most important components (Table 3a,b). Among the hyperparameters,

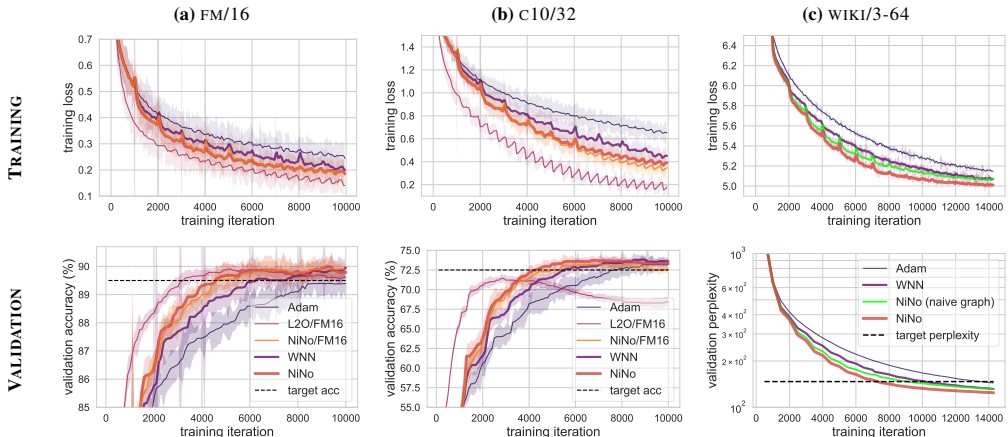

Figure 6: Training loss and validation performance on FashionMNIST, CIFAR-10 and WIKI.

we found that context length $c = 5$ is a sweet spot (Table 3d), because small $c$ may not have enough information to capture the parameter trend, while larger $c$ leads to the NiNo applied too rarely (for $c = 10$ it is applied every $10 \times 200 = 2,000$ steps). Using deeper (Table 3e) and wider (Fig. 5a) GNNs is also important. However we found that depth $M > 3$ or width $D > 128$ make it more challenging to train a performant model. Increasing the number of models ($C$) helps NiNo achieve better speedups (Table 3f, Fig. 5b). In contrast, WNN+ saturates fast when $D$ or $C$ is increased.

**Evaluating on larger models and Llama3-style architectures.** We evaluated the ability of NiNo to speed up training of much larger models than it was trained on. Specifically, we trained 4 and 6 layer GPT2-style Transformers on WIKI with 128 and 384 hidden units having around 7M and 29M parameters respectively (Fig. 5c). The latter is around 18 times larger than the largest in-distribution architecture. Moreover, the WIKI dataset is

Table 4: **WIKI validation perplexity (↓) on a Llama3-style architecture (reported as the mean and standard deviation for 3 runs).**

| Method | 10000 steps | 14000 steps |
|---|---|---|
| AdamW | $26.29{\pm}0.05$ | $24.44{\pm}0.01$ |
| AdamW+NiNo | $\mathbf{24.36{\pm}0.11}$ | $\mathbf{22.37{\pm}0.13}$ |

different from LM1B used in meta-training. Despite these challenges, NiNo was able to speed up training by around 40% and 15% for the 7M and 29M models respectively, outperforming WNN+. We also trained a Llama3-style Transformer with 6 layers and 384 hidden units (111M parameters) with AdamW and AdamW+NiNo, which is an even more challenging setup than in the previous experiments because NiNo did not observe Llama models during its meta-training (Table 1). In this experiment, we use the Llama 3 tokenizer (Dubey et al., 2024) having a larger vocabulary than in GPT2, so we use an ablated NiNo without $\mathbf{V}^w$. Besides the vocabulary, Llama models may pose additional challenges for NiNo, e.g. they differ from GPT2 in parameter initialization, MLP structure, layer normalization, positional embeddings, using biases, grouped query attention, etc. (see Fig. 9 for GPT2 and Llama neural graph examples). Nevertheless, even in this setup our NiNo accelerates training reaching the validation perplexity of the AdamW baseline (24.4) in around 30% less steps (Table 4).

## 5.3 ANALYSIS

**Parameter space ($\theta \to \mathbb{R}^2$).** We show how the parameters evolve during training on C10/32 by projecting them to 2d using PCA (Fig. 7). The PCA projection matrix is computed using the entire Adam trajectory, so the same projection is applied to all the methods. We show the first 8,000 steps for Adam/AdamW and 4,000 steps for other methods with 200 steps between points. In the beginning of training, both WNN and NiNo make big steps (dashed lines in Fig. 7) along the Adam/AdamW

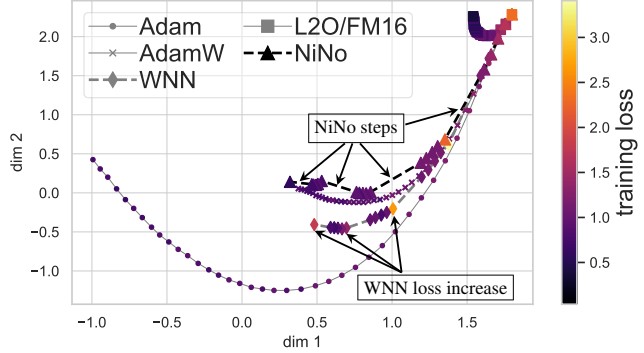

Figure 7: Comparison of optimization methods on the C10/32 task by projecting **parameters** to 2d using PCA at every training step.

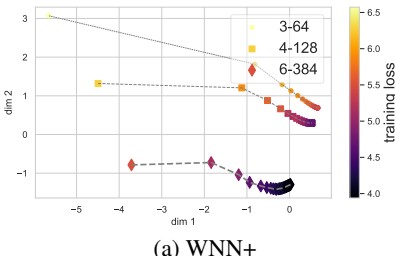 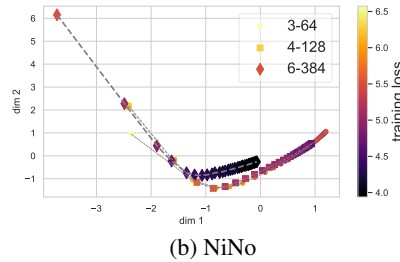

(a) WNN+            (b) NiNo

Figure 8: Comparing the embedding quality of WNN+ (a) and NiNo (b) during optimization by projecting **neural graph embeddings** to 2d using PCA every 1,000 steps. We show the training trajectories of three GPT2-style Transformers with 3, 4 and 6 layers respectively.

trajectory when predicting the parameters, while L2O diverges early. At later optimization stages, NiNo makes better predictions than WNN as the latter leads to significant loss spikes. Interestingly, NiNo's trajectory closely corresponds to a $2\times$ faster AdamW, even though NiNo's base optimizer is Adam in this task. This indicates that NiNo may implicitly regularize optimization.

**Graph embedding space ($\mathcal{E}^M \to \mathbb{R}^2$).** The parameter space analysis used above can be effective on a single task. However, for several tasks with different model sizes ($|\theta|$), finding a common PCA projection matrix ($\theta \to \mathbb{R}^2$), required for the parameter space analysis, becomes challenging. Using neural graphs and NiNo enables such an analysis. Specifically, we can first represent the parameters in the graph embedding space using NiNo and then project the embeddings to 2d. We use this approach to see how the graph embeddings evolve when training three different GPT2-style Transformers on the language task (Fig. 8). We construct $D$-dimensional graph embeddings by computing the average edge features ($1/|\mathcal{E}^M| \sum_{ij} \mathcal{E}^M_{ij}$) after the last NiNo (or WNN+) layer by feeding the 5 past parameter states every 1,000 steps as in our other experiments. Despite having different Transformer architectures, the graph embedding space is the same, which allows us to project the embeddings to 2d using PCA. The PCA projection matrix is computed for each method (NiNo or WNN+) based on the entire trajectory for all the three architectures to allow for visualization on the same plot. Surprisingly, even WNN+ can encode parameter in a meaningful way – in general the parameters of different architectures and at different optimization stages are visually distinct (Fig. 8a). However, WNN+'s embeddings virtually collapse at later training stages for lower loss values (e.g. for 6-384), while NiNo's embeddings are visually more distinct for lower loss values (Fig. 8b). The ability of NiNo to distinguish parameters at later optimization stages can be useful in practice for multiple reasons. For example, despite similar training losses, model parameters can be quite different resulting in different behaviors on downstream tasks (Liu et al., 2023; Dahl et al., 2023).

**Computational costs.** NiNo is applied only every 1,000 steps of the base optimizer, so the wallclock time overhead of using our default NiNo is small, e.g. <0.5% for a 7M Transformer compared to around 5% by L2O (Table 7). As for the memory required for a prediction with NiNo, in our implementation NiNo's peak memory usage is similar to the peak memory usage of a training step of the Transformer with a batch size of 16-32. We expect that in future work, NiNo's inference efficiency can be improved in multiple ways. For example, we can potentially parallelize NiNo's inference on large neural graphs using approaches similar to large scale model inference in image generation (Li et al., 2024).

Additional experiments are presented in Section A.8, the pseudo-code is in Section A.9.

# 6 CONCLUSION AND FUTURE WORK

We proposed NiNo, a novel approach to accelerate training of neural networks, including Transformers. Our work makes a step in the relatively underexplored area of *periodic parameter prediction*, where our experiments show that there is a lot of potential for reducing training time. Our experiments also reveal potentially useful byproducts of NiNo, such as low-dimensional encoding of network parameters during training, which can be used to analyze diverse models and their training dynamics. An interesting direction of future work is to investigate the scaling up of our approach, in particular as the scaling trends that we have observed in our experiments indicate better speedups with more data and larger models.

ACKNOWLEDGMENTS

The experiments were in part enabled by computational resources provided by Calcul Quebec and Compute Canada. Simon Lacoste-Julien is a CIFAR Associate Fellow in the Learning in Machines & Brains program as well as a Canada CIFAR AI Chair. Guillaume Lajoie is a Canada CIFAR AI chair as well a Canada Research Chair in neural computations and interfacing. E.B. and A.M. acknowledge support from Mila-Samsung Research Grant.

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

# A    APPENDIX

## A.1    WNN PARAMETER SCALING DETAILS

In WNNs (Jang et al., 2023), parameter scaling is performed by first computing the minimum and maximum values in the input parameter values: $s_\tau^i = \max(\theta_\tau^i, ..., \theta_{\tau_c}^i) - \min(\theta_\tau^i, ..., \theta_{\tau_c}^i), \forall i \in [1, n]$. Each $i$-th parameters scaling factor $s_\tau^i$ is then used to scale both the inputs: $\tilde{\theta}_\tau^i = \theta_\tau^i / (s_\tau^i + \epsilon)$, and, only for training WNNs, the targets: $\tilde{\theta}_{\tau+k}^i = \theta_{\tau+k}^i / (s_\tau^i + \epsilon)$. Jang et al. (2023) also subtract the last element from the sequence, however we found this to be redundant. Unscaling is performed by multiplying the prediction by $s_\tau^i$. Scaling and unscaling are performed per parameter (for each parameter independently).

## A.2    LINEFIT

Sinha et al. (2017); Jang et al. (2023) introduced a "linefit" baseline to predict future parameter values by extrapolating the line to a future point:

$$\forall i \in [1, n]:$$
$$\hat{\theta}_{\tau+k}^i = 2a^i c + b^i, \tag{10}$$

where $a^i, b^i$ are obtained by optimizing the following objective:

$$\operatorname{argmin}_{a^i, b^i} ||a^i \boldsymbol{x} + b^i - \boldsymbol{\Theta}_{\tau:\tau_c}^i||_2, \tag{11}$$

where $\boldsymbol{x} = [1, 2, ..., c]$ and $\boldsymbol{\Theta}_{\tau:\tau_c}^i = [\theta_\tau^i, \theta_{\tau-1}^i, ..., \theta_{\tau_c}^i] \in \mathbb{R}^c$. This baseline fits a line $(a^i, b^i)$ for each parameter $i$ given only the past values of the same parameter without collecting a training dataset. Despite its simplicity, this baseline was shown to improve on using Adam/SGD only.

## A.3    LINEFIT+

Linefit outlined in Section A.2 can be viewed as a form of momentum that promotes the parameters to be consistent with their global trend. However, Linefit weighs all past values uniformly when fitting the line, while in the momentum more recent values have more effect on the weight update (Sutskever et al., 2013). Motivated by this observation, we introduce the *Linefit+* baseline that penalizes more the errors for later parameter values in the trajectory:

$$\operatorname{argmin}_{a^i, b^i} ||\boldsymbol{\mu}(a^i \boldsymbol{x} + b^i - \boldsymbol{\Theta}_{\tau:\tau_c}^i)||_2, \tag{12}$$
$$\text{where } \boldsymbol{\mu} = [1/c, 2/c, ..., 1].$$

## A.4    NAIVE NEURAL GRAPH OF TRANSFORMERS

A multi-head self-attention (MSA) layer of Transformers (Vaswani et al., 2017) consists of three weight matrices $\mathbf{W}^q, \mathbf{W}^k, \mathbf{W}^v$ applied to the input $\mathbf{x} \in \mathbb{R}^{N \times d}$ and another weight matrix $\mathbf{W}^o$ returning the output, where $N$ is a sequence length and all the four weight matrices are $d \times d$. To compute MSA with $H$ heads, the weight matrices are typically split into $H$ groups across columns for $\mathbf{W}^q, \mathbf{W}^k, \mathbf{W}^v$ and rows for $\mathbf{W}^o$ (Fig. 2a). The heads are processing the input in parallel, $\forall h \in [1, H]$ using Equation 3.

Neural graphs defined by Kofinas et al. (2024) and Lim et al. (2024) are conceptually similar, however Lim et al. (2024) model biases $\mathbf{b}^{(l)}$ and normalization layers as extra nodes connected to the neurons in the $l$-th layer (Fig. 2c). Normalization layers are similarly added as separate nodes, hence the node features of Lim et al. (2024) do not include any parameters. We build on this variant, since it is more easily to implement it for the layers with complicated neuron permutation symmetry, such as multi-head self-attention. Compared to Kofinas et al. (2024), Lim et al. (2024) also provide a slightly different description for the MSA neural graph , however the $\mathbf{W}^q, \mathbf{W}^k, \mathbf{W}^v$ edges are also stacked. Given the lack of detailed description for the multi-head case and no implementation currently available publicly, it makes it challenging to use their neural graphs directly. Therefore, we use (Kofinas et al., 2024) as the base (*naive*) neural graph when constructing MSA layers.

## A.5 GRAPH NEURAL NETWORK

A Graph Neural Network (GNN) layer with edge features can be formulated using the mean aggregation as follows (Corso et al., 2020): $\mathbf{v}_i = \phi_a \left( \frac{1}{|\mathcal{N}(i)|} \sum_{j \in \mathcal{N}(i)} \mathbf{m}_{ij} \right)$, where $\mathbf{m}_{ij}$ is computed using message passing $\phi_m$ given node and edge features $\mathbf{v}_i, \mathbf{v}_j, \mathbf{e}_{ij}$; $\phi_a$ is an MLP; $\mathcal{N}(i)$ are the neighbors of node $i$. Kofinas et al. (2024) modify how $\mathbf{m}_{ij}$ is computed to better incorporate edge features and introduce an edge update step to make better per edge predictions (Table 5). Stacking such layers form a GNN that (1) does not change the size of the input graph and (2) is permutation equivariant. (1) means that for the neural graphs there are $|\theta|' \times d_{\mathbf{E}}$ input edge features and $|\theta|' \times 1$ predictions, where $|\theta|'$ is the number of trainable parameters in the input model plus auxiliary non-trainable parameters (e.g. residual and head connections). (2) means that any permutation of input nodes results in the same permutation of the output nodes and corresponding edges. These properties make GNNs a suitable model to predict future parameters and to work with neural graphs where nodes (neurons) can be permuted in the ways described in Section A.6.

Table 5: **GNN layer comparison.**

| Step | Typical GNN (Corso et al., 2020) | Neural Graph GNN (Kofinas et al., 2024) |
|---|---|---|
| 1. Message passing | $\mathbf{m}_{ij} = \phi_m \left( [\mathbf{v}_i, \mathbf{v}_j, \mathbf{e}_{ij}] \right)$ | $\mathbf{m}_{ij} = \phi_{\texttt{scale}} \left( \mathbf{e}_{ij} \right) \odot \phi_m \left( [\mathbf{v}_i, \mathbf{v}_j] \right) + \phi_{\texttt{shift}} \left( \mathbf{e}_{ij} \right)$ |
| 2. Aggregation | $\mathbf{v}_i = \phi_a \left( 1/|\mathcal{N}(i)| \sum_{j \in \mathcal{N}(i)} \mathbf{m}_{ij} \right)$ | |
| 3. Edge update | $\mathbf{e}_{ij}^{(k+1)} = \mathbf{e}_{ij}^{(k)}$ | $\mathbf{e}_{ij}^{(k+1)} = \phi_e^{(k+1)} \left( \left[ \mathbf{v}_i^{(k)}, \mathbf{e}_{ij}^{(k)}, \mathbf{v}_j^{(k)} \right] \right)$ |

## A.6 NEURON PERMUTATION SYMMETRY

**Permutation symmetry in neural graphs.** Let $\pi^{\text{good}}(\theta)$ be a permutation of neurons in $\theta$ that does not change the network function: $f(\mathbf{x}, \theta) = f(\mathbf{x}, \pi^{\text{good}}(\theta))$. Likewise, let $\pi^{\text{bad}}(\theta)$ be a permutation that changes the function: $f(\mathbf{x}, \theta) \neq f(\mathbf{x}, \pi^{\text{bad}}(\theta))$. Denoting $\cong$ and $\ncong$ as "isomorphic" and "non-isomorphic" operators on graphs respectively, for neural graphs $\mathcal{G}$ the following equations are satisfied under some assumptions (Kofinas et al., 2024): $\mathcal{G}(\theta) \cong \mathcal{G}(\pi^{\text{good}}(\theta))$ and $\mathcal{G}(\theta) \ncong \mathcal{G}(\pi^{\text{bad}}(\theta))$. Here, by definition, the neural graphs are assumed to be constructed correctly, however, achieving or validating that in practice is not trivial for such networks as Transformers.

**Neuron permutation symmetry experiment.** To validate how well our neural graphs correspond to true neuron permutations in Transformers, we run a simple neuron permutation experiment. We use a single Transformer layer with $d = 12$ and $H = 4$, permute rows and columns in its MSA weight matrices $\theta$ using a uniformly sampled permutation $\pi$, and label each permutation as $y_\pi = 1$ for $\pi^{\text{good}}$ or $y_\pi = 0$ for $\pi^{\text{bad}}$ depending if the Transformer output $f(\mathbf{x}, \theta)$ changes or not:

$$y_\pi = \begin{cases} 1, & \text{if} f(\mathbf{x}, \theta) = f(\mathbf{x}, \pi(\theta)) \\ 0, & \text{otherwise.} \end{cases} \tag{13}$$

We generate 1,000 such permutations with about 500 of good and bad permutations. Then we extract a neural graph embedding for each permutation. For correctly constructed neural graphs the embeddings corresponding to $y_\pi = 1$ should be close to the non-permuted $\theta$. To extract embeddings we use a randomly initialized graph neural network with 3 layers and 32 hidden units based on Section A.5. We then visualize the embeddings in 2d (Fig. 2d,e) and color-code each embedding with $y_\pi$. We also train a logistic regression model $\psi$ that takes a graph embedding $\mathbf{h}$ as input and predicts if it is a good or bad permutation: $\hat{y}_\pi = \psi(\mathbf{h})$, and we compute classification accuracy between $\hat{y}_\pi$ and $y_\pi$. We use all 1,000 samples for training and evaluation $\psi$ to estimate how easily can the graph embeddings be separated based on $y_\pi$ (random guess accuracy is around 50%). We repeated the above experiment for the naive and our neural graphs (Fig. 2, d,e). The visualization and classification results indicate that our neural graph construction respects good and bad permutations well, while naive neural graphs confuse them.

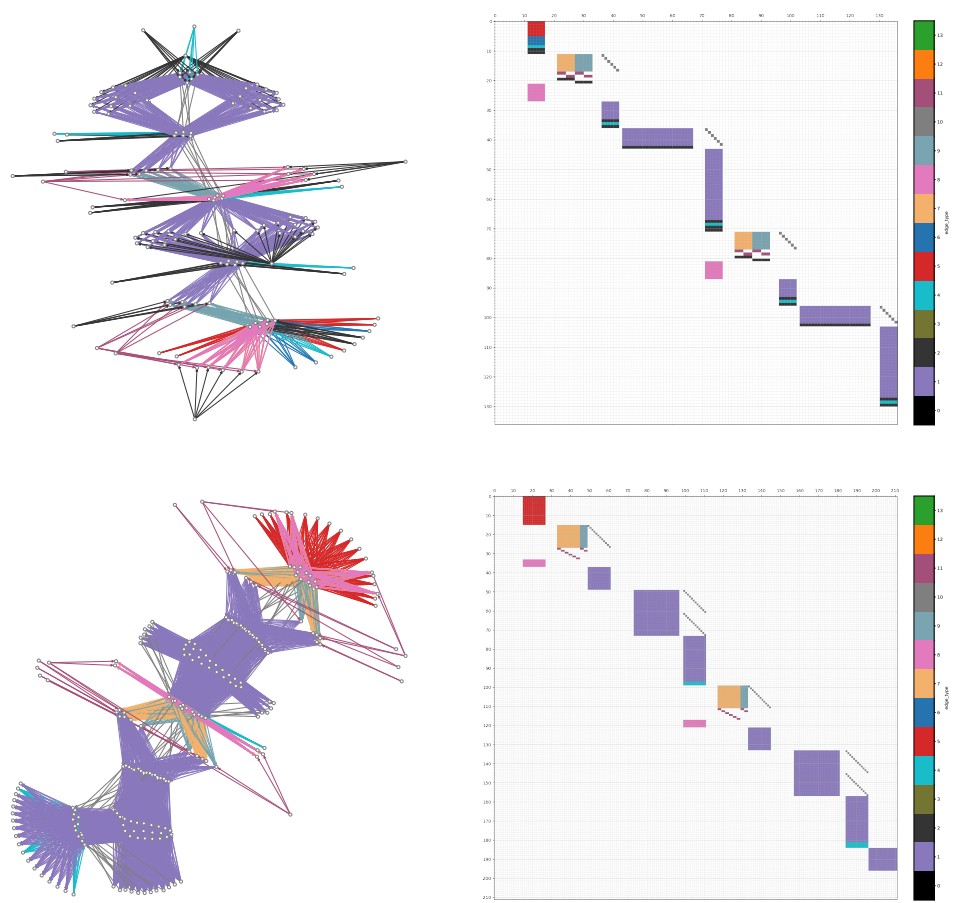

Figure 9: Graph and adjacency matrix with head restrictions and separate q, k, v edge features introduced in Section 4.1, color-coded by edge type. (**top**) A 2 layer GPT2-based Transformer with $d = 6$ and $H = 2$. (**bottom**) A 2 layer Llama3-based Transformer with $d = 12$, intermediate hidden size equal 24, $H = 6$ and number of key value heads equal 2 for GQA. The number of word embeddings is reduced for visualization purposes.

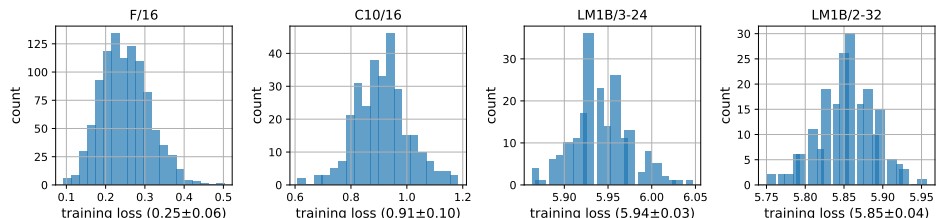

Figure 10: Histogram of final training losses in the in-distribution (meta-training) tasks. See Table 1 for more details.

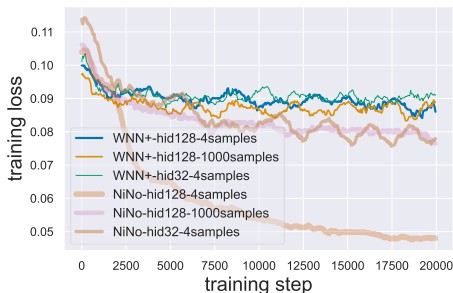

Figure 11: Meta-training curves.

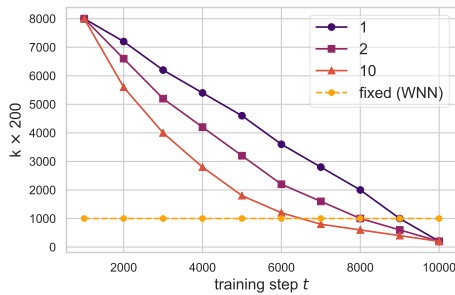

Figure 12: Different $k$-decay schedules.

### A.7 NiNo Training Details

For representing convolutional networks with neural graphs, $d_{\mathbf{E}} = h \times w$ (Kofinas et al., 2024), where $h, w$ are convolution kernel size, so edge features representing $c$ states are $|\boldsymbol{\theta}|' \times chw$, where $|\boldsymbol{\theta}|'$ is the number of trainable and auxiliary non-trainable (residual, head connections) parameters in a model. To make the feature dimensionality the same for different architectures during meta-training, we use zero-padding. We train meta-models for 20k training iterations using AdamW, learning rate 3e-3 with cosine decay and weight decay 0.01. We sample a batch of 4 checkpoints in each training iteration and use automatic mixed precision. Training of the NiNo and WNN+ meta-models completes in under 7 and 6 hours respectively on a single NVIDIA RTX8000 with 48GB of memory.

Meta-training behavior can give important highlights about the model and data. Therefore we explore meta-training for different hidden sizes (128 *vs* 32) and numbers of dataset samples (checkpoints sampled either from all 1000 or just 4 models). We found that NiNo achieves a lower meta-training loss than WNN+ despite having a comparable number of parameters indicating the important of leveraging neural graphs (Fig. 11). However our NiNo model is still in a severe underfitting regime when trained on all the training checkpoints, since when the number of checkpoint models is reduced to just 4 the loss is much lower for NiNo (with 128 hidden units). This result suggests that further scaling up NiNo could help fit all the training data better and potentially achieve stronger results.

### A.8 Additional Experiments

#### A.8.1 Learning rate analysis

When training language models (Section 5.2), we closely followed the setting of WNN (Jang et al., 2023) and used the same learning rate (lr=2e-4). We used this lr for all language tasks, including the collection of training checkpoints (LM1B/3-24 and LM1B/2-32 tasks). Since NiNo was meta-trained on language models trained with lr=2e-4, we use the same lr for all language experiments including Llama3-style (Table 4). However, this lr may be suboptimal for our Llama3-style task, therefore we investigated if our trained NiNo can speed up Adam training with learning rates that are different than those used to collect the checkpoints. We followed the Llama3-style experiment "Evaluating on larger models and Llama3-style architectures" (Section 5.2) with 5 different learning rates run 3 times (Table 6). The standard deviation of the validation perplexity is generally ⩽0.1, indicating that ours is better including the std even in this challenging setup (new lr compared to the training checkpoints, new and 18 times larger architecture and new task). Speedup is computed as described Section 5, e.g. 14% means that AdamW+NiNo achieved the target performance in 12k steps vs 14k steps of AdamW. Based on the results, we conclude that the speedup varies but is significant for different learning rates.

#### A.8.2 ImageNet

To investigate the ability of NiNo on more challenging vision tasks, we trained a small ViT (11M parameters) on ImageNet (Russakovsky et al., 2015), with $32 \times 32$ images as in Metz et al. (2022a); Loshchilov & Hutter (2017). The ViT architecture has 6 transformer blocks with 6 heads and 384 hidden units, the patch size is 2. We trained using AdamW for 50 epochs with learning rate 0.003, cosine scheduler, weight decay 0.3, batch size 3096 and automatic mixed precision using PyTorch and the code base from Knyazev et al. (2023). This setup fully utilizes a single NVIDIA A100-80GB

Table 6: Learning rate analysis for the Llama3-style architecture and the WIKI dataset, extending the results in Table 4. WIKI validation perplexity ($\downarrow$) is reported.

| learning rate | 1e-4 | 2e-4 | 4e-4 | 6e-4 | 8e-4 |
|---|---|---|---|---|---|
| AdamW (10k steps) | 31.85 | 26.29 | 24.81 | 24.50 | 25.03 |
| AdamW + NiNo (10k steps) | **26.18** | **24.36** | **24.10** | **23.82** | **24.49** |
| AdamW (14k steps) - target to compute speed | 28.12 | 24.44 | 23.59 | 23.30 | 23.74 |
| AdamW + NiNo (14k steps) | **24.01** | **22.37** | **22.52** | **22.52** | **22.98** |
| speedup w.r.t. AdamW (14k steps) | 36% | 29% | 21% | 14% | 14% |

training the model in around 6 hours. The setup has around 21k steps in total, with NiNo applied every 1000 steps as in all our experiments. We multiply the predicted parameter delta by 0.1, which we found to be helpful to avoid divergence. Also, to make the NiNo step more efficient and avoid OOM on GPU, we sample 5% of the edges fed to NiNo. The NiNo step takes about 10 seconds, which is similar to 10 steps of Adam, so the overhead is around 1%. After training for 50 epochs, the validation top-1 accuracies are 49.6% (AdamW) vs 50.4% (AdamW+NiNo). So NiNo makes a small yet reasonable improvement given that we are applying the NiNo model to a very different architecture on a very different task compared to meta-training.

Table 7: **Computational cost estimates.** Measured on NVIDIA A100-80GB after running for 1k steps with batch size $b = 32$, sequence length$= 1024$. Note that these numbers are rough estimates as the time/memory vary depending on the GPU/CPU load and internal logic in different PyTorch versions. Also note that time and memory can be traded off by processing only a fraction of parameters at a time. For Adam and L2O, we report per step measurements, since these methods are used at every step. For WNN+ and NiNo, we report per prediction measurements after 1k steps of Adam. "Overhead" denotes wall clock time increase *vs* Adam. [*]The message passing step (Eq. 1-2 in Table 5) of NiNo is computed by sampling 5% of the edges to avoid OOM on GPU.

| METHOD | HID SIZE $D$ | 3-64 (3.4M, GPT2-BASED) | | | 4-128 (7.4M, GPT2-BASED) | | | 6-384 (111.5M, LLAMA3-BASED) | | |
|---|---|---|---|---|---|---|---|---|---|---|
| | | PEAK MEM | TIME | OVERHEAD | PEAK MEM | TIME | OVERHEAD | PEAK MEM | TIME | OVERHEAD |
| Adam | — | 27GB/step | 130ms/step | 0% | 28GB/step | 180ms/step | 0% | 76GB/step | 909ms/step | 0% |
| WNN+ | 32 | 1GB/predict | 172ms/predict | 0.13% | 3GB/predict | 222ms/predict | 0.12% | 42GB/predict | 2.01sec/predict | 0.22% |
| WNN+ | 128 | 4GB/predict | 245ms/predict | 0.19% | 9GB/predict | 294ms/predict | 0.16% | 71GB/predict | 3.44sec/predict | 0.38% |
| NiNo | 32 | 3GB/predict | 323ms/predict | 0.25% | 5GB/predict | 498ms/predict | 0.28% | 57GB/predict | 5.52sec/predict | 0.61% |
| NiNo | 128 | 9GB/predict | 483ms/predict | 0.37% | 16GB/predict | 864ms/predict | 0.48% | 77GB/predict[*] | 13.35sec/predict[*] | 1.47%[*] |
| L2O (mlp) | 32 | 27GB/step | 134ms/step | 3.08% | 28GB/step | 190ms/step | 5.56% | 76GB/step | 917ms/step | 0.88% |

## A.9 PSEUDO-CODE

We show the pseudo-code for the three main steps in our pipeline:

1. collecting the dataset of checkpoints on a set of training tasks;
2. training NiNo given the dataset of checkpoints;
3. evaluating/using the trained NiNo on new tasks.

### A.9.1 COLLECT CHECKPOINTS

```
1  checkpoints = []
2  ckpt_counter = 0   # counter of checkpoints
3  for task in [fm/16, c10/16, lm1b/3-24, lm1b/2-32]:
4    for i in range(300 if task in [fm/16, c10/16] else 200):
5      model = init(task)   # initialize the neural net
6      trajectory = []   # to store the parameters of the model
7      for t in range(10000):
8        model = adam_step(model, task)   # update model parameters using Adam on the task
9        if t % (4 if task in [fm/16, c10/16] else 200) == 0:
10         trajectory.append(model.parameters())   # save only parameters without gradients
11         ckpt_counter += 1
12     checkpoints.append(trajectory)
13
14 print(len(checkpoints))   # 1,000 trajectories (models trained)
15 print(ckpt_counter)   # 1,609,900 total checkpoints
16 print('Done collecting the dataset')
```

### A.9.2 TRAINING NiNo

```
1  nino = NiNo(c=5, gnn_layers=3, K=40)   # define the NiNo architecture (K is for Direct multi-
       step forecasting as described in 4.3)
2  opt = torch.optim.AdamW(nino.parameters(), lr=3e-3, weight_decay=1e-2) # train NiNo with AdamW
3  for step in range(20000):
4    trajectory = sample(checkpoints)   # sample a trajectory of model parameters
5    t = sample(len(trajectory))   # sample some step in the trajectory
6    theta_input = trajectory[t-5:t]   # past 5 parameter states
7    theta_target = trajectory[t:t+40]   # future parameter states
8    theta_input, scales = scale_params(theta_input)   # use our layerwise scaling
9    theta_target = scale_params(theta_target, scales)
10   theta_delta_target = theta_target - theta_input[-1]   # we want to predict the delta w.r.t.
         the last value in the input
11   architecture = architecture(trajectory)   # get model architecture to construct the graph
12   neural_graph = Graph(architecture, edge_features=theta_input)
13   theta_delta_pred = nino(neural_graph)
14   loss = (theta_delta_pred - theta_delta_target).abs().mean()
15   loss.backward()
16   opt.step()   # update NiNo's parameters
17   opt.zero_grad()
```

### A.9.3 USING NiNo

```
1  model = AutoModelForCausalLM.from_config(...)   # some model
2  # NiNo is implemented as a wrapper around the base optimizer
3  # any optimizer other than Adam should also be possible to use with NiNo
4  opt = NiNo(base_opt=torch.optim.AdamW(model.parameters(), lr=1e-3),
5             ckpt='checkpoints/nino.pt',
6             model=model,
7             period=1000,
8             max_train_steps=10000)
9  for step in range(10000):
10   if (step + 1) % 1000 != 0:
11     # Usual Adam updates for most of the steps (e.g. 0-999)
12     opt.zero_grad()
13     data, targets = ...   # get data
14     outputs = model(data)   # forward pass
15     loss = F.cross_entropy(outputs, targets)   # loss
16     loss.backward()   # grads for Adam
17     opt.step(adam=True)   # Adam step or
18     if (step + 1) % 200 == 0:
19       opt.nino_input.append(model.parameters())   # collect the history of parameters
20   else:
21     # no need to compute the gradients for the NiNo forward pass
22     opt.step(adam=False)   # every 1000 steps nowcast params using NiNo
```

