# OpenReview forum: "Accelerating Training with Neuron Interaction and Nowcasting Networks"
_ICLR.cc/2025/Conference — ICLR 2025 Poster_

### Official Review · Reviewer_qcps · 2024-10-31

**Soundness:** 3
**Presentation:** 2
**Contribution:** 3
**Rating:** 5
**Confidence:** 3

**Summary:**

This paper proposes a novel approach to accelerate the training of neural networks by using Neuron Interaction and Prediction (NiNo) networks to predict future parameter changes, thereby reducing the number of iterations required by traditional optimizers such as Adam. This approach shows significant speed-ups on visual and language tasks while maintaining low memory overhead. The theoretical analysis and experimental results of the paper support the effectiveness of this method.

**Strengths:**

The method proposed in the paper has significantly improved compared to the existing technology in multiple benchmark tests, especially in reducing the number of training iterations.

**Weaknesses:**

Hyperparameter tuning: The paper mentions the k-decay strategy, but does not discuss the selection and tuning of hyperparameters in detail. It is recommended that the authors provide more analysis on the impact of hyperparameter tuning on model performance.
Computational resource consumption: Although the paper mentions the memory and time overhead of NiNo, it does not provide a direct comparison with existing methods. It is recommended to add analysis in this regard, especially on resource consumption on large-scale datasets and large models.

**Questions:**

The paper mainly focuses on the performance improvement of specific tasks, but lacks discussion on the generalization ability of the model on unseen tasks. It is recommended to increase the evaluation of the model's generalization ability.

---

> ### Author Response · Authors · 2024-11-12
>
> We thank the reviewer for the time and valuable feedback and would like to promptly respond to the concerns and questions.
>
> > **W1.1**  It is recommended that the authors provide more analysis on the impact of hyperparameter tuning on model performance.
>
> We reported the impact of different hyperparameters (and ablations) in Table 3 and Fig. 5a,b. In Table 3e and Fig. 5a, we show the results for different NiNo's hidden sizes and numbers of layers. In Table 3f, we show the results for different batch sizes, learning rates and numbers of training models (also shown in detail on Fig. 5b). These are the main hyperparameters of NiNo. The ablations in Table 3a,b,c,d complement the analysis.
>
> > **W1.2** Computational resource consumption: Although the paper mentions the memory and time overhead of NiNo, it does not provide a direct comparison with existing methods. It is recommended to add analysis in this regard, especially on resource consumption on large-scale datasets and large models.
>
> In Table 6 in Appendix we compare to the baseline (Adam), WNN+ and L2O. WNN+ has the same memory/overhead as WNN (Jang et al., 2023), since the architecture is identical.
> In addition to Table 6, we recently also measured the memory/overhead of NiNo (with hidden size=32) on our Llama-3-based architecture (with 111M params).
>
> | Method | Peak Mem | Time | Overhead |
> | -------- | ------- | ------- | ------- |
> | Adam | 76GB/step | 909ms/step | 0% |
> | WNN+ (hid=32) | 42GB/predict | 2.01sec/predict | 0.22% |
> | NiNo (hid=32) | 57GB/predict | 5.52sec/predict | 0.61% |
> | L2O (mlp, hid=32) | 76GB/step | 917ms/step | 0.88% |
>
> Overall, NiNo's computational resource consumption is comparable to other approaches, while it consistently outperforms other approaches in terms of speed up.
>
> > **Q1** The paper mainly focuses on the performance improvement of specific tasks, but lacks discussion on the generalization ability of the model on unseen tasks. It is recommended to increase the evaluation of the model's generalization ability.
>
> In our paper we systematically investigated "the generalization ability of the model on unseen tasks". For example, in Tables 1 and 2, we specifically highlight five **Out-of-distribution tasks**. All the five out-of-distribution tasks are unseen: the dataset and/or architecture change compared to the training tasks (FM/16, C10/16, LM1B/3-24, LM1B/2-32). Among the five unseen tasks, the tasks C100/32 and Wiki/3-64 are most different w.r.t. the training ones, because both the datasets and the architectures are unseen. Moreover, in Table 4, we evaluate on the Llama3-based Wiki task, which is the most challenging generalization setup (unseen and very different architecture and unseen dataset).
> We also evaluated a NiNo model trained only on FM-16 (NiNo/FM16) and compared its performance on an unseen task (C10/32) in Fig. 6 (middle). Our NiNo/FM16 outperformed all the baselines (e.g. L2O, WNN).
>
> ==== We will add these clarifications in the revision. We thank again the reviewer and will be glad to clarify further questions.

---

> > ### Author Response · Authors · 2024-11-25
> > **Concerns addressed (hyperparameter tuning and computational resource consumption clarified)**
> >
> > Dear reviewer **qcps**,
> >
> > We would like to politely ask you to review our response, which addresses all the concerns and answers all the questions. The discussion ends soon, on November 26 at 11:59pm AoE.
> > Regarding your Q1 about generalization abilities, we also added a new experiment using a ViT on ImageNet, which is a different architecture and different more challenging dataset than used in meta-training NiNo. See W1.2 in "Part 1 of the response" to reviewer 8oEj for details.
> >
> > Thank you,
> >
> > Authors

---

> > > ### Author Response · Authors · 2024-12-02
> > > **Last day to message the authors**
> > >
> > > We kindly remind reviewer **qcps** that today is the last day that reviewers may post a message to the authors.
> > >
> > > Thank you.

---

### Official Review · Reviewer_PGZt · 2024-11-02

**Soundness:** 3
**Presentation:** 3
**Contribution:** 2
**Rating:** 6
**Confidence:** 3

**Summary:**

The paper introduces NiNo, a method that improves WNN by utilizing graph neural networks (GNNs). Experiments on relatively small datasets demonstrate that NiNo achieves superior performance compared to the WNN baseline.

**Strengths:**

1. The paper is well-motivated. It is an improved version of WNN by integrating GNN.
2. The experiments cover different tasks including language modeling and image classification tasks.

**Weaknesses:**

1. It is unclear if training multiple models during meta-training is practical for real-world applications, where typically only a limited number of models are trained.
2. The generalization performance of NiNo should be further tested. The largest test case is 100 M models on small dataset like Wikitext-103. It may not fully represent NiNo's capabilities in broader applications.

**Questions:**

1. I notice that there are some strange performance jumps in Figure 6. Training loss periodically jumps/spikes for NiNo variants. Could the authors clarify the reasons behind these fluctuations?
2. How is NiNo or WNN applied during inference stage? Specifically, do you forward NiNo k steps and set the network weights to the output of NiNo while retaining the optimizer states? Further details on the inference process would improve clarity.
3. Could the authors provide details about the computational costs of the meta-training process?

---

> ### Author Response · Authors · 2024-11-12
> **Response to concerns and questions**
>
> We thank the reviewer for the time and valuable feedback and would like to promptly respond to the concerns and questions.
>
> > **W1** It is unclear if training multiple models during meta-training is practical for real-world applications, where typically only a limited number of models are trained.
>
> We assume the reviewer is concerned that training 1k models used for meta-training NiNo is not practical. In this regard, in our paper we presented the results of NiNo with a smaller number of models (in the range 40-1000, see Fig. 5b). For example, NiNo's average speed up is around 45% with just 100 models, making NiNo's training practical and performant. So we believe that for real-world applications, it should be possible to train NiNo on existing models without collecting a specific meta-training dataset. We collected such a dataset to enable a systematic study and remove potential confounding factors.
>
> > **W2** The generalization performance of NiNo should be further tested. The largest test case is 100 M models on small dataset like Wikitext-103. It may not fully represent NiNo's capabilities in broader applications.
>
> In our paper (Fig. 5c), we show how NiNo performs when applied to models of different sizes (from 2 to 6 layers). The trend shows that further generalization is unlikely. Nevertheless, the generalization performance is already remarkable in our view because as Fig. 5c highlights (red rectangular), NiNo can be applied to models that are around **18 times larger than those used in meta-training**. So to generalize beyond 100M models, one needs to scale up meta-training models and/or further improve NiNo's architecture. This is left for future work.
>
> > **Q1**  I notice that there are some strange performance jumps in Figure 6. Training loss periodically jumps/spikes for NiNo variants. Could the authors clarify the reasons behind these fluctuations?
>
> When NiNo makes predictions, most of the parameters are generally predicted in the right direction however some parameters may be predicted inaccurately. For the loss spike it can be enough to have just a few parameters to be predicted way off. However, a few training iterations with Adam can easily correct those wrong parameter values. So NiNo makes a good global-level prediction while Adam can fix any local-level mistakes.
>
> > **Q2** How is NiNo or WNN applied during inference stage? Specifically, do you forward NiNo k steps and set the network weights to the output of NiNo while retaining the optimizer states? Further details on the inference process would improve clarity.
>
> The "set the network weights to the output of NiNo while retaining the optimizer states" is correct (we also tried resetting the states but that didn't change the results significantly). Overall the inference process is the following. NiNo is applied every 1k steps of Adam (see Fig. 1). For example, given 10k training steps in some task, NiNo is applied at steps 1k, 2k, ..., 9k, 10k. NiNo's forward pass is done for a specific k value, where k is decayed during the optimization process (Fig. 12 in Appendix). For example, for 10k step the k is decayed as: k=40 at 1k, 33 at 2k, ..., 1 at 10k.
>
> Pseudo code of using NiNo (see supplementary material for full code):
> ```
> model = AutoModelForCausalLM.from_config(...)  # some model
> # NiNo is implemented as a wrapper around the base optimizer
> # any optimizer other than Adam should also be possible to use with NiNo
> opt = NiNo(base_opt=torch.optim.AdamW(model.parameters(), lr=1e-3),
>            ckpt='checkpoints/nino.pt',
>            model=model,
>            period=1000,
>            max_train_steps=10000)
> for step in range(10000):
>     if opt.need_grads:  # True for Adam steps (e.g. 0-999)
>         opt.zero_grad()
>         data, targets = ...  # get data
>         outputs = model(data)  # forward pass
>         loss = F.cross_entropy(outputs, targets)  # loss
>         loss.backward()  # grads for Adam
>     opt.step()  # Adam step or, every 1000 steps, nowcast params using NiNo
>     ...
> ```
>
> > **Q3** Could the authors provide details about the computational costs of the meta-training process?
>
> Training of NiNo completes in under 7 hours on a single NVIDIA RTX8000 with 48GB of memory (see Section A.7 in Appendix). It can be further sped up using multiple GPUs as in standard training pipelines. There are many other potential approaches to make training more efficient (e.g. subgraph/subnetwork sampling common in the GNN works, which can be interesting to investigate in future work). Collecting the meta-training dataset is relatively cheap (can be done in a few hours) and the dataset takes around 200GB (raw memmap data, which we will release upon publication). But as pointed out above, it would be intriguing to try already existing datasets of models in future work.
>
> ====
> We will clarify these concerns in the revision.
> We thank again the reviewer and will be glad to clarify further questions.

---

> > ### Comment · Reviewer_PGZt · 2024-11-26
> >
> > Thank you for your clarification. After reading the responses and reviews from other reviewers, I decide to keep my original rating.

---

> > > ### Author Response · Authors · 2024-12-02
> > >
> > > Thank you for your time and valuable feedback!

---

### Official Review · Reviewer_Ubaj · 2024-11-03

**Soundness:** 3
**Presentation:** 2
**Contribution:** 3
**Rating:** 6
**Confidence:** 3

**Summary:**

This paper introduces a novel approach for accelerating the training of neural networks. This technique called NiNo builds upon the concept of Weight Nowcaster Networks (WNNs), which periodically predict future parameter values thus speeding up optimization. However, in contrast to WNNs, NiNo leverages the inherent network structure by incorporating neuron connectivity using graph neural networks. This allows the model to achieve more accurate prediction of future parameters and leads to faster training. The authors also address challenges in modeling neuron connectivity, particularly in Transformers, and demonstrate NiNo's effectiveness in accelerating Adam optimization in various vision and language tasks.

**Strengths:**

1. The paper is sufficiently well written and is fairly accessible.
2. The proposed approach is sufficiently sound and novel. Even though it can be seen as a combination of two existing techniques (WNNs and an improved GNN model weight representation), this paper still contains a number of non-trivial innovations. For example, among other things, the authors make a number of logical steps improving on a previously published graph topology for multi-headed self-attention.
3. Experimental results appear to be promising. When fully realized the proposed technique could potentially be quite impactful for training modern large models including large language models. Even 10-20% improvements in training speed could be of large practical significance.

**Weaknesses:**

1. Some discussions could perhaps be improved upon to be even more clear. For example, while being sufficiently understandable, Section 4.1 could still be clarified further. Figure 2 is also difficult to interpret in its current form. Color coding takes time to digest.
2. The training method is fairly computationally expensive as the authors collect on the order of $10^6$ checkpoints. To be practical, this initial computational investment should be compensated by the future computational wins from utilizing a more efficient optimizer. Early results seem to paint an optimistic picture and suggest that trained models generalize to much larger underlying models and novel datasets and tasks. However, most current practical LLMs start at around 1B parameters, which still leaves at least 1 to 2 orders of magnitude from the current 111M models the authors experimented with.

**Questions:**

1. How could one bridge the gap between current experiments and practically interesting large model sizes (1-100B parameters)? Would generation of multiple future steps (various values of $k$) present a major obstacle? What about stacked feature representations that gather information from $c$ weight instances?
2. I could have missed this discussion, but what would happen if one decoupled the number of steps between predicting model weights and the size of the history used to make this prediction (both currently chosen to be $c$ if I am not mistaken)? It would seem that there are roughly three time hyper-parameters (at least locally): (a) how frequently do update predictions; (b) how many past states to use for making these predictions (lower would be more advantageous); (c) how far ahead to predict (larger would be more advantageous).

---

> ### Author Response · Authors · 2024-11-18
> **Part 1 of the response**
>
> We thank the reviewer for the time and valuable feedback and would like to respond to the concerns and questions.
>
> > **W1** Some discussions could perhaps be improved upon to be even more clear. For example, while being sufficiently understandable, Section 4.1 could still be clarified further. Figure 2 is also difficult to interpret in its current form. Color coding takes time to digest.
>
> We just updated the pdf to make Figure 2 more clear (increased the contrast and augmented the notation with different styles). We will be glad to further improve it according to feedback.
>
> To clarify Section 4.1, below we are rephrasing the two problems of naive neural graphs (Kofinas et al., 2024) that we are addressing:
> 1. in naive neural graphs it’s possible to have neuron/node permutations that **do not change the graph** (i.e. it remains isomorphic to the original graph), but **change the transformer output**. Such an example would be swapping neuron 3 (belongs to head 1) with neuron 4 (belongs to head 2). We address this by allowing the permutations **only within the head**.
> 2. in naive neural graphs it’s also possible to have neuron permutations that **do not change the transformer output**, but **change the graph**. Such an example would be permuting neurons in $W^q$ and $W^k$ while not permuting neurons in $W^v$. We address this by assigning separate edges for $W^q,W^k$ and $W^v$ instead of having 3-dimensional (for q,k,v) edges, so in our case the neurons in $W^q, W^k$ can be shuffled independently from $W^v$ according to the underlying function of MSA layers.
>
> As we show in our synthetic experiments in 4.1 (illustrated in Figure 2d,e and explained in Appendix A.6), our modifications indeed better correspond to the correct neuron symmetry in MSA layers. That is in our neural graphs, when the graph does not change, then the transformer output does not change either, and vice versa: if the graph changes, the transformer output changes as well.
>
> We will be glad to further clarify this or other pieces.
>
> > **W2** The training method is fairly computationally expensive as the authors collect on the order of $10^6$
>  checkpoints.
>
> While the total number of checkpoints is $10^6$, the total number of models we trained is only $10^3$. Moreover, in our paper we also presented the results of NiNo with a smaller number of models (in the range 40-1000, see Fig. 5b). For example, NiNo's average speed up is around 45% with just 100 models, making NiNo's training practical and performant. So we believe that for real-world applications, it should be possible to train NiNo on existing models without collecting a specific meta-training dataset. We collected such a dataset to enable a systematic study and remove potential confounding factors. We are answering to "1B parameters" below in Q1 (see **Part 2 of the response**).

---

> > ### Author Response · Authors · 2024-11-18
> > **Part 2 of the response**
> >
> > > **Q1.1** How could one bridge the gap between current experiments and practically interesting large model sizes (1-100B parameters)?
> >
> > - **Larger training checkpoints.** As we show in Fig. 5c (red rectangular), NiNo can generalize to models that are around **18 times larger than those used in meta-training**. So one straightforward approach to bridge the gap would be to train NiNo on 100M-1B models. To do so without expensive training of these checkpoints, leveraging existing trained models would is possible (since intermediate checkpoints are also often available, e.g. [1,2]).
> >
> > - **Efficient GNN layers.** For scalability, it is also important to make NiNo more efficient at training and inference. We implemented graphs and the GNN layers based on Kofinas et al. which were applied to models with only around 0.1M params, so it is not implemented efficiently. There are many works in the GNN literature to scale graph message passing even to 100M nodes [3,4], so it's definitely possible. For example, the neural graph for a 70B Llama model would have around 10M nodes. While possible, it's logical to leave it for future/follow-up work as it requires a separate study.
> >
> > - **Normalizing/scaling parameters.** Without scaling the parameters the average speed up drops from 48.9% to 32.1% (Table 3a). We think it's possible to further improve our normalization, e.g. [5] proposed Maximal Update Parametrization, which we can potentially leverage for improved generalization.
> >
> > **References**
> > - [1] https://huggingface.co/TinyLlama/tinyLlama-intermediate-checkpoints
> > - [2] https://huggingface.co/BAAI/CCI3-HQ-Intermediate-Checkpoints
> > - [3] https://ogb.stanford.edu/docs/nodeprop/#ogbn-papers100M
> > - [4] SGFormer: Simplifying and Empowering Transformers for Large-Graph Representations, NeurIPS 2023
> > - [5] Tensor Programs V: Tuning Large Neural Networks via Zero-Shot Hyperparameter Transfer, NeurIPS 2021
> >
> > > **Q1.2** Would generation of multiple future steps (various values of $k$) present a major obstacle? What about stacked feature representations that gather information from $c$ weight instances?
> >
> > - Generation of multiple future steps does not cause efficiency issues at inference, because at inference NiNo predicts only for a single $k$ value, which is chosen based on our proposed scheduling (L306 "Inference with k-decay." and Fig. 12). Inference for a single $k$ value is very efficient because the last NiNo layer is a simple linear layer with $K$ outputs (i.e. $y=xW$ where $x \in \mathbb{R}^d$, $W \in \mathbb{R}^{d \times K}$ and $y \in \mathbb{R}^K$). So at inference, we can index the $k$-th column of $W$ before the matrix product like $y_k=xW_k$ to get the predicted parameter value at $k$.
> > - Multiple ($c$) weight instances are also not the main bottleneck, because $c=5$ works well and it can be potentially even further reduced. The main bottleneck is usually the first linear layer that projects $c$ to $d$, where $d$ is for example 128 (in our default NiNo). So for large models we can only process part of the input weights at once (e.g. only first layer) which becomes tricky for the GNN layers which normally require the full graph as input (at least in simple implementations). Therefore, our current evaluation is limited to 100M models.
> >
> > > **Q2** What would happen if one decoupled the number of steps between predicting model weights and the size of the history used to make this prediction (both currently chosen to be $c$  if I am not mistaken)? It would seem that there are roughly three time hyper-parameters (at least locally): (a) how frequently do update predictions; (b) how many past states to use for making these predictions (lower would be more advantageous); (c) how far ahead to predict (larger would be more advantageous).
> >
> > Absolutely, this is a very good point. The WNN paper coupled $c$ and $k$, so at any step $t$ they would always take the past 1000 states as input (with 200 steps apart, so $c=5$) and predict the parameters at step $t+1000$. In our work, we found that predicting further ahead is beneficial, especially in the beginning. About the three hyperparameters:
> > - While **(c) how far ahead to predict** can be treated as a hyperparameter, we wanted to avoid introducing additional hyperparameters and proposed to decay $k$ (see L306 "Inference with k-decay." and Fig. 12), which worked well across different tasks.
> > - **(b) how many past states**, which is $c$, was discussed above in **Q1.2**.
> > - **(a) how frequently do update predictions** is another hyperparameter we fixed to 1000 steps across all the tasks both during training and evaluation of NiNo. It's definitely possible to tune it, but again we wanted to simplify the usage of our model. We believe a very interesting future direction is to automatically (e.g. by another network) determine at which step to apply NiNo.
> >
> > ==== We will clarify these concerns in the revision. We thank again the reviewer and will be glad to clarify further questions.

---

> > > ### Comment · Reviewer_Ubaj · 2024-11-25
> > >
> > > I would like to thank the authors for a very thorough response! It addressed all my questions and provided valuable insights into the potential applications of this technique within large language models (LLMs). While acknowledging that it's beyond the current scope, I remain wondering about the scalability of this technique to significantly larger models (which might be a limiting factor in its practical implementation). Nonetheless, it will be exciting to see how this or similar approaches could lead to the development of new and accelerated training methods.

---

> > > > ### Author Response · Authors · 2024-12-02
> > > >
> > > > Thank you for your time and positive feedback!

---

### Official Review · Reviewer_8oEj · 2024-11-04

**Soundness:** 2
**Presentation:** 2
**Contribution:** 3
**Rating:** 6
**Confidence:** 3

**Summary:**

The paper proposes to improve training, by leveraging out specific transformer strcture and representing it is as a Neural Graph to model connectivity. The weights of the model would then be encoded as features on the edges of the graph (the nodes corespond to neuron/activation).  Ideally,  isomorphic permutations of such graphs,
should result in no-functional change to the model itself, thus capturing the underlying symmetries of the model. In contrast with earlier work, authors proposed a more fine-grained structure that accurately captures multi-headed-self-attention blocks, and thus allows them to improve performance of their meta-learning method. Once such graph is built authors leverage a combination of nodes and edge embedding, methods from GNN which are then translated to model updates..

**Strengths:**

I think the construction of the graph whose isomorphic permutations of nodes preserves the model functionality,  for transformer layer is neat and interesting.

**Weaknesses:**

I have two major concerns about the papers:
1. My main concern is that the experimental section contains only fairly trivial datasets (FashionMNist, Cifar-10), which are very far from anything reasonable these days, and the models authors consider for forecasting is limited to ~1M parameters, and many are 15K params,  which is barely practical for the simplest tasks. I think for image tasks, showing reasonable performance on something like ImageNet is
a must.  On the other hand authors run their training procedures for thousands of steps, which seems an overkill  for simple problems like this (Gradient Descent for Fashion Mnist for instance can converge in ~100 steps).  On the other hand experiments on Liama are not convincing because it is a well  known fact that it is often easy to speed up initial convergence, but the end result can still be significantly worse than slower algorithm. To demonstrate performance gains on these tasks authors should run at least run ablation study with different learning rates of the original model and show that they all converge (hopefully their method still shows the improvement).

2) The actual algorithm descriptions has very little detail what exactly their meta-learning algorithm does, even when reading the  appendix  carefully. Most of the detail is in section 4.2. In particular having at least a basic definition of GNN layers. Given that it is the crux of the paper, i think it should be carefully expanded and explained. I think the graph-construction section, is fairly well written, but it is very much unclear how it is then translated into actual meta-learning algorithms, as best as i could tell authors use some black-box message passing algorithms to come up with the new update, but i think that section would benefit greatly from expansion.

3) Some discussion about optimality of their graph construction would be nice, but it is a minor comment.

**Questions:**

1. How are  the gradients are fed into the NiNo algorihtm? It is part of V^w?
2. What are the nodes features?
3. Was there any ablation study done for AdamW in Liama3 style architecture particularly around using different learning rates.

---

> ### Author Response · Authors · 2024-11-19
> **Part 1 of the response**
>
> We thank the reviewer for the time and valuable feedback and provide our response below.
>
> **High-level response:** Please note that with academic-level GPU resources it is often very challenging to scale up the experiments of very novel methods to industry-level standards (>=1B parameters).
> Our work has made significant and non-trivial contributions in terms of novelty compared to the very recent works, WNN (Jang et al., ICML 2023) and neural graphs (Kofinas et al., ICLR 2024). Regarding the scale, Kofinas et al. applied neural graphs to the models with only ~0.1M parameters, so we have significantly (**by 3 orders of magnitude**) scaled up the neural graph approach when applied it to a Llama3-style transformer with 111M params (see our Table 4). This kind of scale is also common in optimization literature [1, 2, 3], and scaling up is often studied in follow up works.
> At the same time, we thank again the reviewer for valuable feedback that helps to improve our paper, which we updated. Please see the detailed response below.
>
> > **W1.1** Experimental section contains only fairly trivial datasets (FashionMNist, Cifar-10), which are very far from anything reasonable these days, and the models authors consider for forecasting is limited to ~1M parameters, and many are 15K params.
>
> For Llama3-style experiments the NiNo is applied to a 111M parameter model, also the results in Figure 5c are for GPT2-based models with up to 30M params, so it is much larger than ~1M parameters.
> Our experiments are in-line with recent literature on experimental optimization procedures, e.g. [1, 2].
> We thus believe it is reasonable to leave further scaling up to future work as it requires a separate study. For example, one key direction to investigate could be developing more scalable GNN layers.
>
> > **W1.2** For image tasks, showing reasonable performance on something like ImageNet is a must.
>
> We trained a small ViT (11M parameters) on ImageNet (with 32$\times$32 images as in [2, 3], for 50 epochs given the short response time). The validation top-1 accuracies are 49.6% (Adam) vs 50.4% (Adam+NiNo), which we believe is a reasonable improvement on ImageNet, especially given that we are applying the NiNo model to a very different architecture on a very different task compared to meta-training. We will add this experiment to the revision.
>
> > **W1.3** Authors should at least run ablation study with different learning rates of the original model and show that they all converge (hopefully their method still shows the improvement).
>
> We attempted to stick closely to the setting of the recent work WNN. Therefore we used the same learning rate (lr=2e-4) as in WNN. We used this lr for all language tasks, including the collection of training checkpoints (lm1b/3-24 and lm1b/2-32 tasks). Since NiNo was meta-trained on models trained with lr=2e-4, we use the same lr for all experiments including Llama3-based.
> As in WNN we limit our analysis to lr used in checkpoints, but using a different lr than those from checkpoints at inference time is an interesting question. We have now done a study for different learning rates (without retraining NiNo on these new lr checkpoints):
>
> | learning rate | 1e-4 | 2e-4 | 4e-4 | 6e-4 | 8e-4 |
> | ------ | ------ | ------ | ------ | ------ | ------ |
> AdamW (10k steps) | 31.85 | 26.29 | 24.81 | 24.50 |  25.03 |
> AdamW + NiNo (10k steps) | **26.18**  | **24.36** | **24.10** | **23.82** | **24.49** |
> AdamW (14k steps) - target | 28.12 | 24.44 | 23.59 | 23.30 |  23.74 |
> AdamW + NiNo (14k steps) | **24.01** | **22.37** | **22.52** | **22.52** | **22.98** |
> speed up w.r.t. AdamW (14k steps) | 36% | 29% | 21% | 14% | 14% |
>
> We ran each experiment 3 times, the standard deviation is generally <=0.1, indicating that ours is better including the std even in this challenging setup (**new lr compared to the training checkpoints, new and 18 times larger architecture and new task**).
> Speed up is computed as described in our paper (e.g. 14\% here means that AdamW + NiNo achieved the target performance in 12k steps vs 14k steps of AdamW). As reviewer **Ubaj** noted, even 10-20\% improvements in training speed could be of large practical significance. We will add this analysis to the revision.
>
> **References:**
> - [1] Learning to Generalize Provably in Learning to Optimize, AISTATS 2023
> - [2] Practical tradeoffs between memory, compute, and performance in learned optimizers, CoLLA 2022
> - [3] Decoupled Weight Decay Regularization, ICLR 2019

---

> > ### Author Response · Authors · 2024-11-19
> > **Part 2 of the response**
> >
> > > **W1.4** Authors run their training procedures for thousands of steps, which seems an overkill for simple problems like this (Gradient Descent for Fashion Mnist for instance can converge in ~100 steps).
> >
> > Indeed, with a very large batch size, it may be possible to converge in 100 steps. However, a common well-studied setting in the FashionMNIST/CIFAR kind of tasks is batch size=128 (see optimization literature [1, 2, 3] cited in the Part 1 of the response, as well the WNN and neural graph papers we are based on). So just observing all data on these datasets once requires around 400 steps.
> > With 100 steps and the same batch size, the model would observe only 1/4 of the data, which is not sufficient in this setting. As we show in Table 1 in our submission, well-tuned Adam requires several thousand steps to achieve the target performance on our tasks.
> >
> > > **W2.1** Having at least a basic definition of GNN layers. Given that it is the crux of the paper, i think it should be carefully expanded and explained.
> >
> > Indeed, we added a short description in Section 4.3 (L292-296) of the submission pdf to make it more clear. On a high level, a GNN layer takes node and edge features as input, performs neighborhood aggregation operations and edge feature updates, and outputs updated node and edge features.
> >
> > > **W2.2 (related to Q1 below)** The actual algorithm descriptions has very little detail what exactly their meta-learning algorithm does, even when reading the appendix carefully.
> >
> > Given a related question about the gradients as input (**Q1**), we would like to emphasize that our NiNo model is trained with a simple supervised loss **between the predicted parameters** and **the parameters obtained by Adam** (see eq 1 and 9). To train NiNo we do not need the access to data (images, text), we only need the checkpoints of model parameters $\theta$ (**without gradients**, see Figure 4, which we slightly updated in the pdf to improve its clarity).
> > We use the terms meta-training and meta-dataset when talking about NiNo to differentiate between training the NiNo model (i.e. its parameters $\phi$) and training task specific neural networks ($\theta$).
> > However, no learning-to-learn or meta-learning is performed. In meta-learning the loss to train NiNo would be a task loss (e.g. cross-entropy in image classification), so during training the NiNo model the predicted parameters would be evaluated on the mini-batch of the task data and backpropagated to the NiNo model. This is often done in learning to optimize. As we discussed in the Introduction and Experiments (L422), our approach is much simpler than l2o in terms of training, has less overhead in total and generalizes much better.
> > Our idea is based on the WNN paper, so after the NiNo model is trained, it is used as the following: we use Adam for 1k number of steps, then apply the NiNo model to predict future parameters, then run Adam for another 1k steps, apply NiNo and so on (see Fig. 1). The NiNo prediction step is based  on the past parameters (without gradients), the neural graph and, implicitly, the training dataset of checkpoints.
> >
> > **To improve our paper given this feedback, we updated the pdf and included page 19 in the Appendix showing pseudo-code** for the three main steps in our pipeline: (1) collecting the dataset of checkpoints on a set of training tasks; (2) training NiNo given the dataset of checkpoints; (3) evaluating/using the trained NiNo on new tasks.
> >
> > > **W2.3** Some discussion about optimality of their graph construction would be nice, but it is a minor comment.
> >
> > We evaluated the graph construction quality in the synthetic neuron permutation experiment visualized in Fig 2d,e and described in Appendix (A.6). To summarize, we validated how well our neural graphs correspond to
> > true neuron permutations in Transformers. We use a single Transformer layer with d=12 and H=4, permute rows and columns in its MSA weight matrices $\theta$ using a uniformly sampled permutation, and label each permutation as $y$=1 for
> > good permutation (does not change the transformer output $f(x, \theta)$) or $y$=0 for bad permutation (changes transformer output). We generate 1,000 such permutations of MSA layers with about 500 of good and bad permutations. Then we
> > extract a neural graph embedding for each permutation. For correctly constructed neural graphs the
> > embeddings corresponding to $y$=1 should be close to the non-permuted $\theta$. We also train a logistic regression model to quantitatively estimate how well the embeddings are clustered according to $y$, We repeated the above experiment for the naive and our neural graphs (Fig. 2, d,e). The visualization and classification results indicate that our neural graph construction respects good and bad permutations well, while naive neural graphs confuse them.

---

> > > ### Author Response · Authors · 2024-11-19
> > > **Part 3 of the response**
> > >
> > > > **Q1** How are the gradients are fed into the NiNo algorihtm? It is part of $V^w$?
> > >
> > > Our NiNo model does not take gradients as input (we updated Figure 4's caption to make it more clear and added pseudo-code in Appendix, page 19). NiNo is trained by a simple supervised loss following the WNN paper (described in Section 3.1), which only takes the history of parameters as input. Please see the response to **W2.2** above for more details.
> > >
> > > > **Q2** What are the nodes features?
> > >
> > > We described **Node features** in L243 in the submission. To highlight this paragraph better, we updated the pdf and made a separate Section "4.2 Edge and Node Features".
> > > Specifically, we use Laplacian Positional Encoding (LPE) computed based on 8 smallest non-trivial eigenvectors of the neural graph adjacency matrix. So each node has 8 features. For Transformer’s word embedding layers, we also found it beneficial to leverage a positional feature (the number from 1 to 50257, set to 0 for all other layers). LPE and this positional feature are first projects to $d$-dimensional features using embedding layers and then summed up (see our supplementary material with implementation for details), so each node becomes $d$-dimensional. Note that our NiNo models are trained only on the GPT2 models, so for the Llama3-based experiments we use the NiNo variant without the positional feature because Llama3 and GPT2 have different tokenizers.
> > >
> > > > **Q3** Was there any ablation study done for AdamW in Liama3 style architecture particularly around using different learning rates.
> > >
> > > Yes, please see the response to **W1.3** above.
> > >
> > > ==== We conclude our response here. We thank again the reviewer for improving the paper and will be glad to clarify further questions.

---

> > > > ### Author Response · Authors · 2024-11-25
> > > > **Concerns addressed (more experiments, meta-training explained)**
> > > >
> > > > Dear reviewer **8oEj**,
> > > >
> > > > We would like to politely ask you to review our response, which addresses all the concerns and answers all the questions. The discussion ends soon, on November 26 at 11:59pm AoE.
> > > > We especially highlight the clarification of the meta-training algorithm (see Part 2 of the response and page 19 in Appendix of the submission where we added the pseudo-code).
> > > >
> > > > Thank you,
> > > >
> > > > Authors

---

> > > > > ### Comment · Reviewer_8oEj · 2024-11-26
> > > > >
> > > > > I appreciate detailed response with explanations and new experiments,
> > > > > i think with these new experiments in mind, i will raise the score to 6
> > > > >  (my concern still remains about the scalability of this approach, but it
> > > > >  seems promising enough to warrant the publication)

---

> > > > > > ### Author Response · Authors · 2024-12-02
> > > > > >
> > > > > > Thank you for your time and positive feedback!

---

### Author Response · Authors · 2024-11-22
**Thank the reviewers and reminder about the discussion deadline on Nov 26th**

We thank the reviewers a lot for providing very valuable reviews, which helped us to improve the paper. We would like to kindly ask the reviewers to review our responses and updated pdf submission given that the discussion ends soon, on November 26 at 11:59pm AoE. We will be happy to address any remaining concerns.

---

### Meta-Review · Area_Chair_o5V4 · 2024-12-21

**Metareview:**

This paper proposes a method to accelerate the training of neural networks called NiNo, which uses neuron connectivity to better predict the near-future parameters, via a graph meta-network, during the optimisation process. Overall, the reviewers found the construction of the graph network to be interesting, and that the conducted experiments appear to be promising. The main weakness is the scalability of the approach, and how effective the approach would be beneficial to large-scale models, as the validation focuses on relatively simple datasets. The AC also agrees with this point.

**Additional Comments On Reviewer Discussion:**

During the rebuttal period, the authors provided additional experiments demonstrating the approach's potential to generalize to larger models. While the reviewers raised their rating, they still have concerns regarding the scalability of this approach. The AC agrees with the scalability concerns. Overall, the strengths seem to outweigh the weaknesses by a bit, and the AC recommends the acceptance of this paper.

---

### Decision · Program_Chairs · 2025-01-22

Accept (Poster)